# Temporal transcriptional regulation of mitochondrial morphology primes activity-dependent circuit connectivity

Iryna Mohylyak[1], Maheva Andriatsilavo [1], Mercedes Bengochea[1], Carlos Pascual-Caro [1], Noemi Asfogo[1], Sara Fonseca-Topp [1], Natasha Danda[1], Marlene Cassar[1], Corentine Marie[1], Zeynep Kalender Atak[2], Maxime De Waegeneer[2], Stein Aerts[2], Olga Corti[1], Jaime de Juan-Sanz[1] & Bassem A. Hassan [1] ✉

Synaptic connectivity during development is known to require rapid local regulation of axonal organelles. Whether this fundamental and conserved aspect of neuronal cell biology is orchestrated by a dedicated developmental program is unknown. We hypothesized that developmental transcription factors regulate critical parameters of organelle structure and function which contribute to circuit wiring. We combined cell type-specific transcriptomics with a genetic screen to discover such factors. We identified *Drosophila* CG7101, which we rename <u>mi</u>tochondrial integrity <u>r</u>egul<u>a</u>tor of <u>n</u>euronal <u>a</u>rchitecture (*Mirana*), as a temporal developmental regulator of neuronal mitochondrial quality control genes, including *Pink1*. Remarkably, a brief developmental downregulation of either Mirana or Pink1 suffices to cause long-lasting changes in mitochondrial morphology and abrogates neuronal connectivity which can be rescued by *Pink1* expression. We show that Mirana has functional homology to the mammalian transcription factor TZAP whose loss leads to changes in mitochondrial function and reduced neurotransmitter release in hippocampal neurons. Our findings establish temporal developmental transcriptional regulation of mitochondrial morphology as a prerequisite for the priming and maintenance of activity-dependent synaptic connectivity.

To form functional circuits, neurons deploy a plethora of cell biological processes. During axon growth and synapse formation the regulation of organelles, such as vesicles and mitochondria, plays crucial roles in the development of functional synaptic connectivity[1–3]. These organelles are known to rapidly sense and react to changes in the environment of a developing axon to ensure appropriate responses over time. In contrast, it is unknown what role, if any, developmental transcription factors in the nucleus play in regulating axonal organelle biology to coordinate robust circuit development and function.

Mitochondria are important and highly dynamic organelles within the neuron, which are present both in the neuronal soma and neurites. Mitochondrial function is tightly regulated by structural dynamics, and they form distinct subpopulations within neuronal compartments in response to specific cellular demands[4–7]. In mature neurons, mitochondria are involved in synaptic transmission and plasticity regulation through local ATP supply and $Ca^{2+}$ buffering[6,7]. During development, neurite outgrowth and branching are directly affected by the proper transport, dynamics, and anchoring of mitochondria[4,8,9].

¹Institut du Cerveau-Paris Brain Institute (ICM), Sorbonne Université, Inserm, CNRS, Hôpital Pitié-Salpêtrière, Paris, France. ²Laboratory of Computational Biology, Center for Brain and Disease Research – VIB-KU Leuven, Leuven, Belgium. ✉e-mail: bassem.hassan@icm-institute.org

In addition to their own genome, mitochondria depend crucially on genes encoded by the nuclear genome that are essential for their biogenesis and function, in response to cellular energy and growth requirements. Due to their importance in regulating processes such as energy production, calcium buffering, and cell survival, mitochondrial quality is ensured by multiple systems of quality control[10]. Proper maintenance of the axonal mitochondrial pool is now widely thought to be important for brain health[11]. Mitochondrial dynamics are important for axonal targeting, synaptogenesis, and circuit maintenance. Their metabolic activity sets the species-specific tempo of neuronal development and is proposed to affect the dynamics of transcription during brain development[12]. However, despite their outsized impact on brain development, relatively little is known about the active transcriptional regulation of mitostasis at different steps of neuronal growth and connectivity.

The fly visual system, with its well-described developmental programs of cell patterning and circuit wiring, has been a powerful tool to dissect the molecular mechanisms of neuronal circuit formation and to extract the fundamental rules that these mechanisms subserve. Both retinal neurons and higher order visual system neurons, such as the Dorsal Cluster Neurons (DCNs/LC14s) have been used to identify genetic hierarchies that lead to the emergence of neuronal circuit wiring specificity[13,14]. Using the DCNs as a model system we report the discovery of a temporally expressed transcription factor we named Mirana (mitochondrial integrity regulator of neural architecture), required for axonal growth and activity-dependent circuit connectivity. We show that Mirana is a developmental transcriptional regulator of key mitochondrial quality control genes. Developmental loss of Mirana suffices to induce strong and long-lasting enlargement of axonal mitochondria. These changes can be rescued upon re-expression of its target genes, notably the Parkinson's disease gene Pink1, demonstrating a causal link. Thus, early developmental transcriptional regulation of mitostasis is required for priming future neuronal activity and circuit connectivity.

## Results

### DCNs transcriptome analysis

DCNs are a cluster of 22–68 cells per hemisphere marked by their dorsal location and expression of the transcription factor Atonal (Ato). They are divided into 2 subtypes based on their axonal target choice: a minority of DCNs innervate the contralateral medulla neuropile and are called M-DCNs (Fig. 1A), while the majority innervate the lobula neuropile and are called L-DCNs[15] (Fig. 1A'). As a rare and highly variable cell type, DCNs tend to not be highly represented in single-cell RNAseq data[16] and cannot be unambiguously distinguished from other Ato+ neurons, such as the Ventral Cluster Neurons (VCNs)[17]. Because the DCNs are an example of a rare cell type in the fly brain (-120 cells/brain), it was challenging to ascertain their full and specific transcriptome from single-cell RNAseq datasets where typically a minority of DCN cells are represented. Similarly, other techniques, like TRAP (Translating Ribosome Affinity Purification) or TaDa (Targeted expression of the DamID) rely on using a cell-type specific Gal4 driver, not expressed anywhere else. To specifically profile the DCN transcriptome, we used Laser Micro Dissection, followed by bulk next-generation RNA sequencing (RNA-SMARTseq2)[18] of isolated DCN clusters. We generated a range of microdissected DCNs samples and random cuts in central brain regions of the same animals as control samples (Fig. 1B). After MDS clustering analysis of the top 1000 variable genes, we retained 6 bulk samples of DCNs and 2 control samples for further investigation (Supplementary File 1, Supplementary Table 1). The molecular identity of isolated DCN clusters was confirmed by strong expression of the marker gene ato[17], absent in control samples, and decreased levels of main glial markers (repo, moody, wraper, alrm, Eaat2 and Gat). Housekeeping genes were expressed at similar levels (Fig. 1C). We used a range of threshold parameters (logFC >2, adjusted p-value ≤ 0,01, a mean base number ≥100) and manual analysis of available gene annotations to filter the list of candidate genes for further analysis (Fig. 1D, Supplementary File 1, Supplementary Tables 1 and 2). Nevertheless, as it was extremely challenging to combine LMD with subsequent FACS sorting, we cannot exclude possibility of some level of contamination from neighboring cell types.

We reasoned that some of the DCN-expressed genes might encode factors required for regulation of target genes that in turn control local axonal mechanisms during circuit assembly. If so, it should be possible to identify such factors through a genetic screen for DCN connectivity defects. To test this idea, we performed a developmental RNAi-screen of 74 significantly enriched genes in DCNs and analyzed changes in DCN wiring pattern. As a first filter, we scored the number of DCN medulla axons and set a phenotypic threshold of more than 30% change in the mean number of M-DCN axons with a p-value lower than 0.01 compared to control flies (Supplementary Fig. S1). This resulted in 10 candidate genes whose downregulation decreased, increased, or completely impaired medulla innervation, some in a sexually dimorphic manner (Fig. 1E–H). Among those were transcription factors (CG7101), genes involved in chromatin remodeling (Brd8, polybromo, Hira), cytoskeleton organization (Cnb), and actin filament dynamics (ssh). Here we report the functional analysis of the previously uncharacterized role of the transcription factor encoded by CG7101 in neural circuit development.

### CG7101 downregulation causes DCNs Medulla innervation loss

We identified CG7101, a gene encoding a putative zinc-finger transcription factor, which we rename Mirana (mitochondrial integrity regulator of neuronal architecture) based on our functional characterization (see below). Given that the RNAi screen reveals a requirement in developing DCNs, we queried developmental single-cell RNA sequencing data[19] and found that Mirana mRNA is moderately expressed in developing DCNs at 15 h after puparium formation (P15) and is then progressively downregulated throughout pupal development, before being re-expressed at very low levels in adult flies (Supplementary Fig. S2A). This is also true across neuronal cell types in the developing brain (Supplementary Fig. S2B). Next, we compared our list of the highest confidence upregulated genes in LMD dissected DCNs to scRNAseq data set generated by Ozel et al.[19] in cluster 27_LC14, that partially corresponds to DCNs, and found that the majority of genes were shared between the two lists (Supplementary Fig. S3). Using a transgenic line carrying GFP-tagged Mirana (67655[BDRC]) to examine the developmental expression profile of Mirana protein in DCNs, we found that expression peaks at P30–P40, followed by downregulation at P50 and very low levels of expression in DCN cells in adults (Fig. 2A, A'-G, G'). Next, we confirmed the loss of medulla axons phenotype observed in the initial screen that was detectable already at the late pupal stage using two independent RNAi lines (#100127, #27849) and CRISPR/Cas9 approach[20], by expressing a Mirana gRNA and Cas9 specifically in DCNs (Fig. 2H–M). This phenotype was not degenerative, as we did not detect differences between M-DCN axons number in young and old flies in both control and Mirana-RNAi background (Supplementary Fig. S2C). These findings strongly indicated that Mirana acts as temporal developmental transcription factor, within a specific developmental time window sensitive to its expression levels. To determine if this is the case, we combined expression of Mirana-RNAi transgene in DCNs with temperature sensitive tub-Gal80[ts] repressor. We used this strategy to downregulate Mirana expression at two sequential time points, from L3 till P48 APF (detected peak expression) and from P48 to adults (no or very low expression levels). We confirmed, that Mirana knockdown at the time of its peak expression leads to decreased medulla innervation (Fig. 2N–O, R), in contrast to downregulation during later stages, which had no effect on medulla

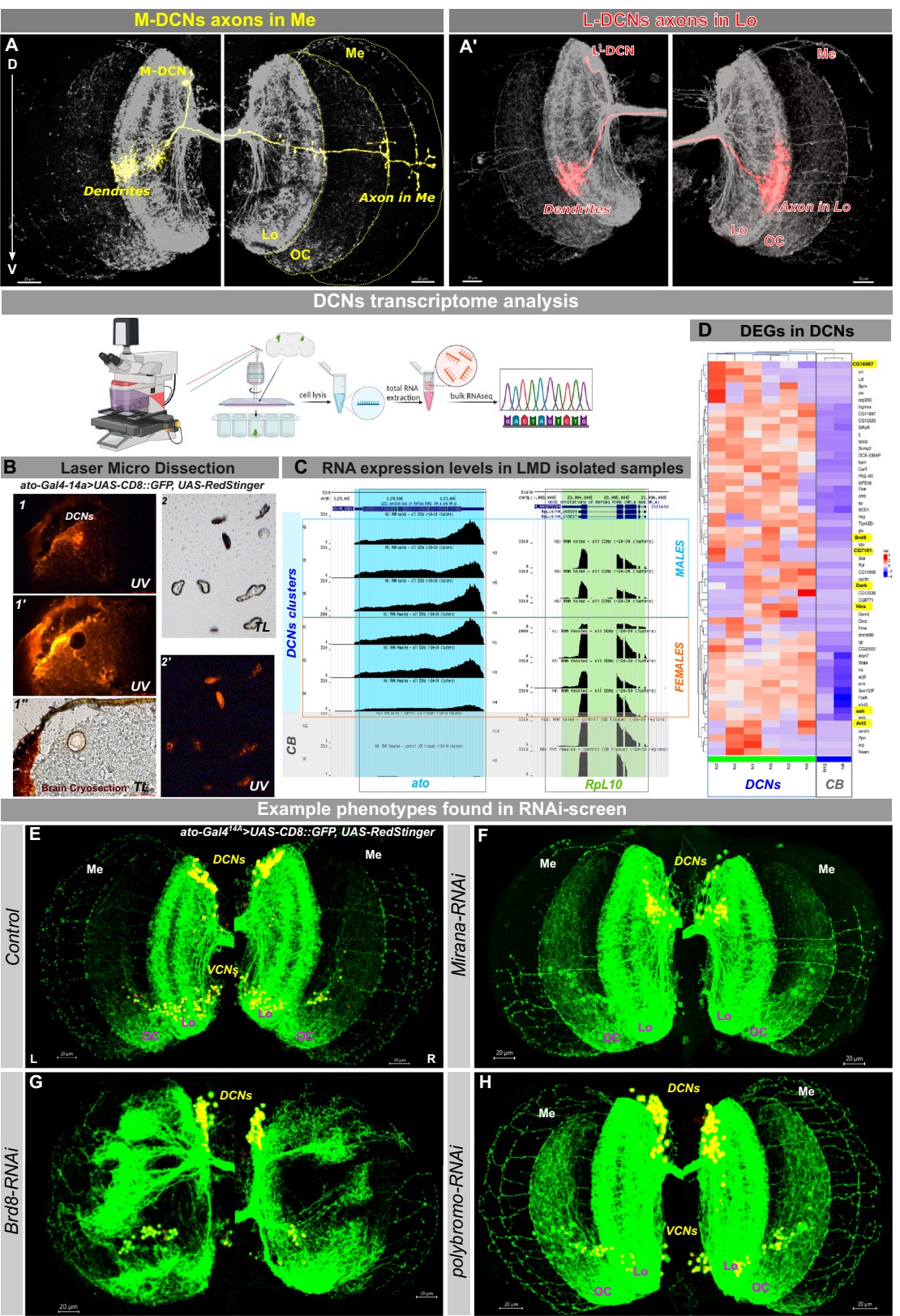

axons (Fig. 2P–R). This time window coincides with the time when DCNs cross the optic chiasm from the lobula to the medulla (at P48)[21].To identify the developmental origin of this decrease we examined DCN axons at this stage and the number of axons is equal between control and *Mirana* knock down DCNs, suggesting that Mirana is required for maintaining DCN axons in the medulla after they have crossed the chiasm. To identify the fate of the retracted

axons, we generated stochastic single cell clones and asked whether any of them showed an abnormal targeting outside either the medulla or lobula and found this not to be the case (Supplementary Fig. S4A–D). We conclude that Mirana encoded by *CG7101*, is a developmentally regulated transcription factor specifically required at early stages of pupal development for the maintenance of distal DCN axons in their medulla target region.

**Fig. 1 | DCNs transcriptome analysis with subsequent RNAi screen of 74 upregulated genes. A**, **A'** Multi Color Flip-Out (MCFO) staining of single DCN cell innervating Medulla (M-DCN) and Lobula (L-DCN). **B** Laser Micro Dissection of DCN clusters from brain cryosections of *ato-Gal4^{14A};UAS-CD8::GFP,UAS-RedStinger* adult flies. 1, 1'- RedStinger fluorescent DCNs before (1, UV light) and after (1', UV light, 1'', transmitted light) dissection. 2, 2' - sample with dissected DCNs clusters in the transmitted (2) and UV (2') light. Total RNA was extracted using Ambion RNAqueous Micro Kit and libraries were generated using SMART-seq2 Kit and Nextera Tagmentation Kit. NGS NextSeq 500 High Output Kit (400 million reads of apr. 25 million reads/sample coverage was used for sequencing). Created in BioRender.

Hassan, B. (2025) https://BioRender.com/fma66dn. **C** Example of the marker gene *atonal* high-level expression in DCN samples compared to random central brain regions used as a control, with the equal expression of housekeeping genes. **D** Heatmap of top-20 genes expressed in DCNs. **E–H** RNAi-screen of 74 DCN-enriched genes revealed 10 gene candidates causing significant phenotypical changes upon DCN-specific downregulation during development. Lo lobula, OC optic chiasm, Me medulla, DCNs Dorsal Cluster Neurons, VCNs Ventral Cluster Neurons. Scale bar – 20 μm. Genes expressed in DCNs and list of genes for the RNAi-screen are listed in Supplementary Fig. S1. Results of RNAi screen of 74 genes in Supplementary Fig. S1.

## Mirana and human TZAP bind to and regulate mitochondrial biology genes

Mirana shows a degree of homology to a few of mammalian zinc finger genes, including human TZAP/ZBTB48 (33% similarity and 24% identity mostly in the zinc finger domains, Supplementary Fig. S5B), PRDM5 (36% similarity and 20% identity, Supplementary Fig. S5C) and ZNF497 (34% similarity and 26% identity, Supplementary Fig. S5D). TZAP/ZBTB48 is a zinc finger transcriptional factor, which is composed of an N-terminal BTB/POZ domain and eleven adjacent C2H2-type zinc fingers (Znf1-11) at its C-terminus[22]. TZAP/ZBTB48 is mainly known for its role in telomere length control in the context of cancer biology[23,24] but telomere length control is regulated differently in flies and humans. A previous study reported that human TZAP activates the expression of the *Mitochondria Fission Process 1 gene (MTFP1)* potentially affecting mitochondrial morphology. We wondered whether fly Mirana might also regulate mitochondrial biology genes. We analyzed available CG7101/Mirana ChIPseq data from the ModERN project[25] and TZAP ChIPseq data from human cell lines generated in the previous study of human TZAP[26]. Our analyses reveal that, in addition to *MTFP1*, TZAP shows binding peaks in genes encoding other mitochondrial regulatory proteins including *DNM1L, PINK1, ULK1, NRF1*, and *MFN2* (Supplementary Fig. S6A, Supplementary Table 2). Similarly, we found that Mirana has binding peaks to the promoter regions of the fly homologs of these human genes involved in mitochondrial morphology, namely *CG7772/dMtfp1, Drp1, Pink1, Atg1, ewg*, and *Marf1* (Supplementary Fig. S6B). To test if TZAP and Mirana regulate these genes, we examined their relative mRNA levels upon TZAP downregulation in mammalian COS7 cells using si-RNA or post-mitotic pan-neuronal knock-out of *Mirana* in fly brains (*snyb-Gal4>Mirana_sgRNA,UAS-U^M-Cas9^{340007}*). This resulted in the loss of 80–90% of *Mirana* expression and significant downregulation (70–80% in vivo) of the 6 mitochondrial biology genes tested in mammalian cells and in fly neurons (Supplementary Fig. S6C, D). These genes are expressed in DCNs during development, with increasing expression starting around P48, following the Mirana protein peak expression timepoint, Supplementary Fig. S6F). Interestingly, overexpression of *Mirana* did not upregulate its target genes (Supplementary Fig. S6E). Thus, Mirana is required for the initiation of developmental upregulation of mitochondrial biology genes in neurons.

## Mirana is a conserved regulator of mitochondrial morphology

Mitochondria are crucial organelles for energy production, regulation of cell signaling, and apoptosis, with their function tightly associated with their morphology[27,28]. We expressed GFP-tagged mitochondrial localization sequence (*UAS-mito-GFP*) in *Drosophila* DCNs and examined mitochondrial morphology in axonal terminals. Downregulation of *Mirana* in DCNs led to a significant increase in mitochondrial size in *Mirana-RNAi^{100127}* and *Mirana_sgRNA, UAS-UM-Cas9* expressing flies (Fig. 3A–D). This mitochondrial elongation phenotype is of developmental origin, as it was detected already in P72 pupae, with a gradual increase until eclosion (Fig. 3E–G), which coincides with the period of active synaptogenesis and neuronal activity in the fly visual system. To ascertain the specificity of the developmental origin of this phenotype we again combined expression *Mirana-RNAi* transgene in DCNs with

the temperature sensitive *tub-Gal80^{ts}* repressor. We observed the same mitochondrial elongation in flies with developmental-specific *Mirana* downregulation (Fig. 3H, I, K), but not in flies with *Mirana* downregulation exclusively in adult flies (Fig. 3J, K). More specifically, knock-down of *Mirana* during its peak expression (L3-P48) is sufficient to cause elongated mitochondria (Fig. 3L, M, O), while *Mirana* downregulation during subsequent developmental stages had no effect on mitochondria size (Fig. 3N, O). To test whether this effect is common to all DCN subtypes and cellular compartments, we combined the dendritic marker DenMark[29] with mito-GFP, and examined mitochondrial size in DCN dendrites, and L-DCN axons. As in of M-DCN axons, we observed a significant elongation in both compartments (Supplementary Fig. S4G–J). These observations confirm the developmental requirement of Mirana for proper mitochondrial morphology and axonal targeting.

Mitochondria are dynamic organelles, and their shape and size are precisely controlled by fusion and fission events, which should be balanced for maintaining overall mitochondrial morphology[30,31]. To investigate mitochondrial morphology at higher resolution we examined the effects of TZAP silencing in the COS7 cells expressing mito::GFP, which are well-suited for due to their large and flat cytoplasm and well-developed mitochondrial network. Under control conditions, most cells (60%) contain a mix of mitochondrial shapes, while 20% of the cells show predominantly fused filamentous mitochondria, and another 20% show "donut-like" mitochondria, indicative of cellular stress (Fig. 4A, A'). Following transfection with TZAP siRNA there was a significant increase in the proportion of cells with donut-like mitochondria (from 22% to 50%), a tendency towards more cells with fused filamentous mitochondria (from 17% to 33%), and a concomitant decrease in the proportion of cells with mixed mitochondrial morphologies, compared to the control siRNA condition (Fig. 4A, A'). In addition, in some severe cases within siRNA-TZAP transfected cells, we observed the formation of "blob-shaped" mitochondria (Fig. 4B, C) indicating irreversible toxicity[28]. Quantitative image analysis of mitochondrial morphology in individual cells with a predominantly donut-like mitochondrial network revealed an increase in the mean values of aspect ratio and form factor following *TZAP* silencing compared to control cells, two parameters reflecting the length and the degree of branching of the mitochondrial objects. In addition, the mean values of the perimeter and area of mitochondrial objects were increased, and circularity was decreased in cells exposed to TZAP siRNA compared to cells treated with Control siRNA (Fig. 4D).

Next, we asked what the impact of TZAP loss on mammalian neuronal mitochondrial morphology might be. We expressed a mitochondria-localized $Ca^{2+}$ sensor (mito^{4x}-GCaMP6f)[32] in sparsely transfected rat primary hippocampal neurons, which allowed us to identify single mitochondria in axons and evaluate their morphology using the baseline mito^{4x}-GCaMP6f (Fig. 4E–G). We thus evaluated the area, perimeter, circularity, and aspect ratio and found that *TZAP_shRNA* neurons presented larger mitochondria with a more elongated shape, confirming our observations in mammalian COS7 cells and *Drosophila* neurons.

Altogether, our results so far show that *Mirana* is a conserved developmentally regulated transcription factor that is required to

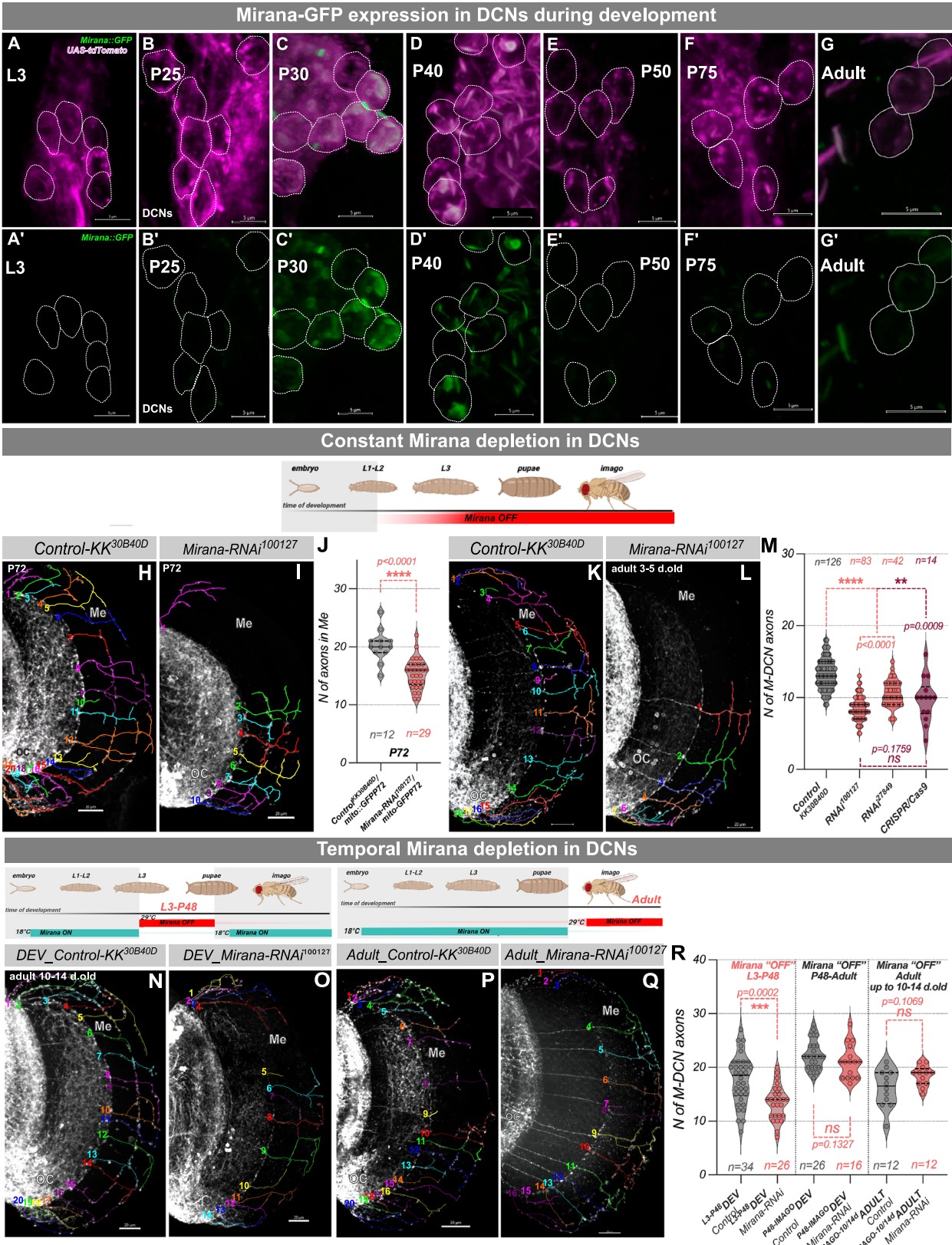

maintain normal mitochondrial size through the regulation of expression of several genes that regulate mitochondrial morphology.

### TZAP downregulation impairs presynaptic glutamate release
We hypothesized that elongated mitochondria could impair neuronal activity. To test whether developmental TZAP expression is required

for presynaptic function in mammalian neurons, we expressed a surface-localized glutamate sensor, iGluSnFR3 in primary hippocampal neurons, in the presence or absence of an shRNA against *TZAP* (Fig. 5H). iGluSnFR3 exhibits fast non-saturating activation dynamics and reports presynaptic glutamate release with specificity[33]. To achieve developmental knock down of TZAP we transfected TZAP

**Fig. 2 | CG7101/Mirana downregulation causes significant reduction of DCN axons in medulla. A–G′** Temporal expression pattern of the tagged Mirana protein (*676SS^BDSC* combined with *ato-Gal4^14A*,*UAS-CD4::tdTomato* transgenes) during development DCNs shows highest peaks of protein detection at 30–40 h after puparium formation (P30-P40 APF) with subsequent decrease, scale bar – 5 μm. **H–L** Constitutive downregulation of *CG7101/Mirana* using two independent RNAi lines (*100127^VDRC* and *27849^VDRC*) and CRISPR/Cas9 approach (*Mirana_sgRNA^341305* and *UAS-U^M-Cas9^34007* from Heidelberg CFD CRISPR library) in DCNs (driven by *ato-Gal4^14A*) leads to a significant decrease in the number of DCNs axons in the medulla, detected at the late pupal stage (P72, **H–J**) and adult (**K–M**). **N–R** Temporal depletion of *Mirana* in DCNs using *Mirana-RNAi^100127*;*ato-Gal4^14A*,*UAS-CD4::tdTomato/tub-Gal80^ts* flies. **N, O** *Mirana* depletion in L3-P48 developmental window is sufficient to recapitulate loss of medulla innervation (**R**). Flies were kept at 18 °C until wondering L3 larvae, and then transferred for 48 h to 29 °C for heat shock

inactivation of Gal80 repressor. After heat shock flies were transferred back to 18 °C and were kept there until 14 days old adults. In contrast, *Mirana* depletion in later developmental stages (from P48 until eclosion) did not cause loss of in medulla innervation (**P–R**). Flies were kept at 18 °C until P48, and then transferred to 29 °C for heat shock inactivation of Gal80 repressor and kept there until eclosion. Adult flies were transferred into 18 °C and were kept there until 14 days old adults before dissection. OC optic chiasm, Me medulla, scale bar – 20 μm. We examined 12 P72 Control optic lobes, 29 P72 *Mirana-RNAi^100127*, 126 optic lobes of control group adult flies (males and females), 83 of *RNAi^100127*, 42 of *RNAi^27849*, 14 of CRISPR/Cas9. In temporal depletion experiment in development: 34 optic lobes of control flies, 26 of *Mirana-RNAi^100127*, in adult – 12 optic lobes of control flies and 12 of *Mirana-RNAi^100127*. Statistical analysis was done using two-tailed Mann–Whitney test. Exact *P* values, *$p < 0.05$, **$p \leq 0.01$, ***$p \leq 0.001$, ****$p \leq 0.0001$, ns not significant. Schematics created in BioRender. Hassan, B. (2025) https://BioRender.com/fma66dn.

siRNA into immature neurons (DIV7) and measured glutamate release two weeks later (DIV 21) We observed that upon *TZAP* knock-down, a higher percentage of neurons failed to show any quantifiable response in comparison to controls (Fig. 5I). Single action potential firing generated robust presynaptic iGluSnFR3 signals (Fig. 5J, K; black trace) as previously reported[33]. Among responding neurons, we observed significantly reduced glutamate release by ~50% in TZAPshRNA-expressing neurons, indicating that in absence of TZAP presynaptic neurotransmitter release is impaired (Fig. 5K; red trace, K′). Thus, TZAP is required in hippocampal neurons for normal mitochondrial morphology, efficient neuronal activity and neurotransmitter release.

During neurotransmission, axonal mitochondria transiently sequester Ca$^{2+}$ in their matrix. Although this buffering may influence vesicle cycling under high-frequency stimulation[34,35], under physiological conditions mitochondrial Ca$^{2+}$ uptake serves as a feedforward mechanism that transiently boosts mitochondrial metabolism, increasing ATP production to compensate for energy expenditure[32,36]. We asked whether the absence of TZAP could modulate mitochondrial Ca$^{2+}$ uptake and consequently alter ATP dynamics during activity. We co-expressed mito$^{4x}$-GCaMP6f with TZAP shRNA and compared axonal mitochondrial Ca$^{2+}$ fluxes in wildtype and TZAP KD neurons during 20AP evoked at 20 Hz using field stimulation (Fig. 5L). We found that impairing TZAP expression caused a ~50% reduction in mitochondrial Ca$^{2+}$ uptake without altering mitochondrial extrusion rates (Fig. 5L). Previous work showed that reduced mitochondrial Ca$^{2+}$ uptake results imbalanced ATP production/consumption, leading to reduced ATP levels after neuronal activity[32]. Using the latest cytosolic ATP sensor, iATPSnFR2[37], we compared ATP dynamics in wildtype and TZAP KD neurons. We found that neurons with reduced TZAP expression presented a more pronounced relative decrease in ATP levels after stimulation (Fig. 5M), suggesting that activity-driven mitochondrial ATP production is less efficient in TZAP KD neurons.

## Loss of Mirana leads to loss of activity-dependent synaptic connectivity and corresponding behavioral defects

To examine the role of Mirana in synaptic connectivity and neuronal circuit activity in vivo we investigated synapse formation and function of *Drosophila* DCN neurons. Although loss of Mirana reduces the number of DCN medulla targeting axons, 40–50% of the expected DCN axons still target the medulla and appear to have normal terminal morphology. We quantified the presynaptic markers BrpD3, Syd1, and Nrx1, by expression of GFP-tagged versions and found no qualitative or quantitative differences between *Mirana-RNAi* and control DCNs (Supplementary Fig. S7A–C). Thus, loss of Mirana does not appear to impair the formation of presynaptic sites. As we observed reduced neurotransmitter release from presynaptic sites in hippocampal neurons upon TZAP depletion, we measured spontaneous synaptic vesicles release in M-DCN axonal terminals by performing live imaging of a dual-tagged pH sensitive sensor pHluorin, fused to stable mCherry fluorophore and Synaptotagmin (Syt1-mCherry::pHluorin)[38]. This

sensor allows the localization of presynaptic release sites with the pH resistant mCherry, while simultaneously permitting quantification of synaptic vesicle exocytosis with the pH-sensitive GFP moiety, which will fluoresce upon vesicle release exposing GFP to the neutral pH of the extracellular environment. We observed a dramatic decrease in GFP pHluorin signal normalized to the mCherry signal in case of *Mirana* downregulation suggesting significant reduction in synaptic vesicle release in DCN axons after Mirana depletion (Fig. 5A-A″, B-B″, C).

To test whether this loss of activity impaired neuronal circuit connectivity, we used the anterograde transsynaptic tracing approach TransTango[39] and quantified the number of positively labeled postsynaptic cells in the medulla (Fig. 5D–F, magenta). This revealed a ~10-fold loss of postsynaptic connectivity with target neurons upon *Mirana* downregulation with both RNAi and CRISPR/Cas9 approaches (Fig. 5D-E″, I, Supplementary Fig. S7D-D′, G, white arrowheads). Consistent with our previous results, we observed loss of TransTango labeling of postsynaptic cells under conditions of *Mirana* knock down uniquely during developmental stages (L1-eclosion, Fig. 5F-F′, I). This reduction did not lead to a change in targeting specificity, because the few cells that did label in Mirana loss of function conditions corresponded to the previously identified postsynaptic targets of DCNs, like Lawf1/2, Tm2/21, DCN, and Dm3/6 neurons[40]. Thus, Mirana is required for presynaptic vesicle release and neuronal circuit connectivity.

The loss of TransTango signal was somewhat surprising because TransTango is not thought to be an activity-dependent method, although this has never been specifically tested to our knowledge. To test whether this is the case, we blocked DCN activity using three different silencers: inhibiting synaptic vesicle exocytosis via Tetanus toxin (*UAS-TNT*)[41], or suppressing electrical activity by expression of potassium channels (Kir2.1 or dORK) that are open at resting membrane potential which results in increased potassium efflux and membrane hyperpolarization setting resting membrane potential below the threshold required to fire action potentials[42]. In all cases, we found the same drastic decrease of postsynaptic labeling observed in the *Mirana* downregulation condition (Fig. 5H-H′, Supplementary Fig. S7E–G). Importantly, expression of the nonconductive form of dORK as a control[43] showed postsynaptic labeling comparable to control flies (Fig. 5G-G′, I).

To ascertain that the loss of TransTango postsynaptic labeling reflects loss of functional connectivity, we tested DCN behavioral function in Mirana loss of function flies. It was previously shown that DCN medulla targeting is required for object orientation behavior of freely walking flies, measured by absolute stripe deviation in the Buridan paradigm assay[44]. Briefly, flies walk freely between two opposite and identical high-contrast stripes which they are unable to reach. The width of the path they trace during these back-and-forth walks is known as absolute stripe deviation (Fig. 5J). We have previously shown that this parameter depends on DCN function and correlates with the

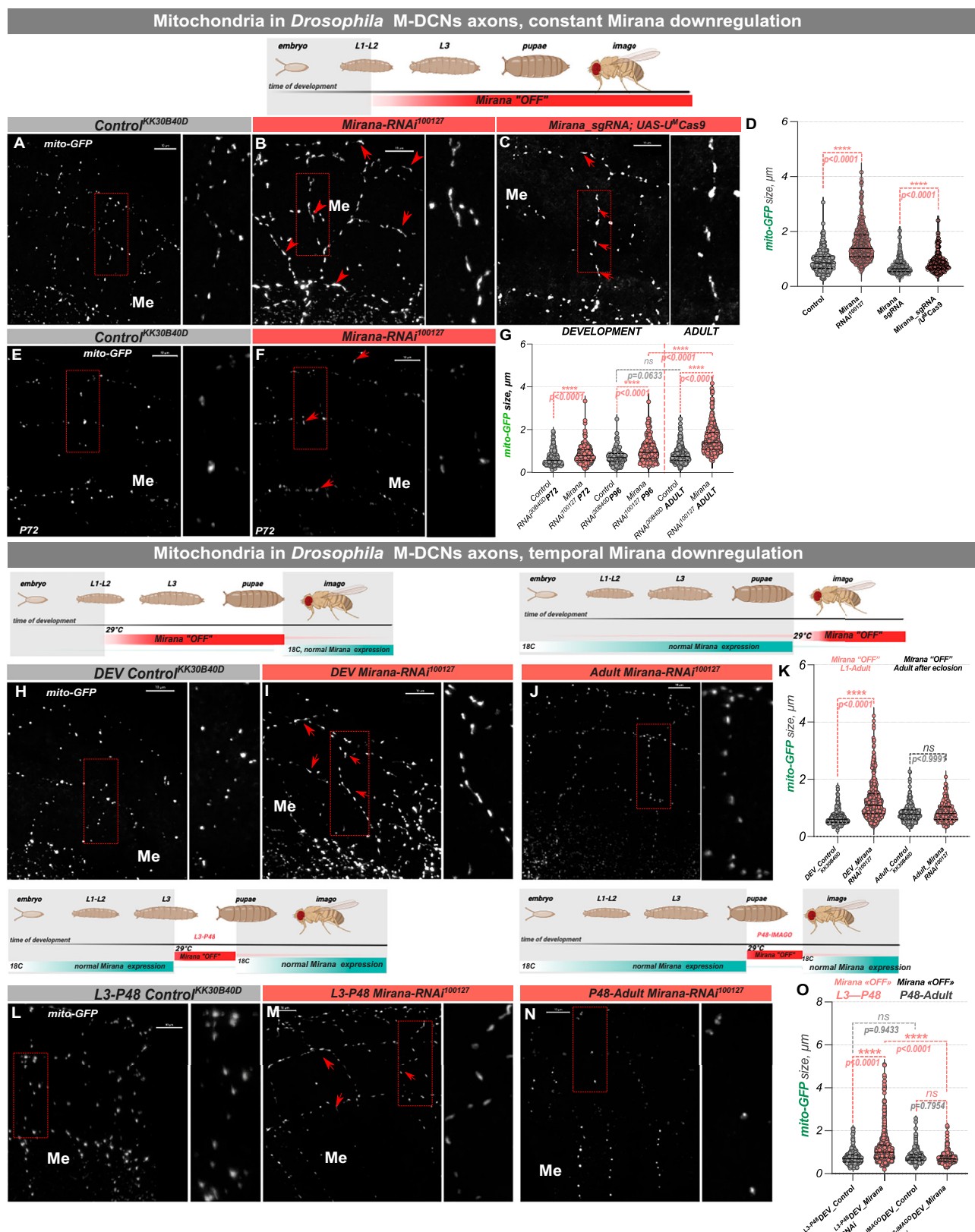

degree of left-right asymmetry in medulla DCNs wiring[45]. Silencing of DCN activity results in a significant increase of absolute stripe deviation at the population level because flies tend to walk across the entire arena and along the edges (Fig. 5K,K′-L,L′). It also results in loss of the correlation between DCN connectivity and behavior in individual flies. Thus, absolute stripe deviation provides a quantitative parameter to

measure DCN circuit function in vivo. We measured absolute stripe deviation in control and DCN-specific Mirana loss function conditions and observed a loss of the correlation between wiring and behavior in individuals (Supplementary Fig. S8A–E) and an increase in absolute stripe deviation (Fig. 5M) quantitatively similar to that observed upon UAS-TNT expression in DCNs[45].

**Fig. 3 | Temporal Mirana depletion in Drosophila neurons leads to increase in mitochondrial size. A–G** Mitochondria in M-DCNs axonal terminals labeled with mito-GFP in Control condition (**A** *w¹¹¹⁸;UAS-mito::GFP/UAS-KK³⁰ᴮ⁴⁰ᴰ;ato-Gal4¹⁴ᴬ,UAS-CD4::tdTomato*) and constant *Mirana* downregulation using RNAi against *Mirana* (**B** *w¹¹¹⁸;UAS-mito::GFP/UAS-Mirana¹⁰⁰¹²⁷;ato-Gal4¹⁴ᴬ,UAS-CD4::tdTomato*) and CRISPR/Cas9 system (**C** *w¹¹¹⁸;UAS-mito::GFP/UAS-Mirana_gRNA;ato-Gal4¹⁴ᴬ,UAS-CD4::tdTomato/UAS-Uᴹ Cas9³⁴⁰⁰⁰⁷*). Elongated mitochondria can be detected already in late pupal stages, and gradually increase in size until the adult stage (**E–G**). **H–O** Temporal *Mirana* downregulation during development (*Mirana-RNAi¹⁰⁰¹²⁷/UAS-mito::GFP;ato-Gal4¹⁴ᴬ,UAS-CD4::tdTomato/tub-Gal80ᵗˢ* vs *UAS-KK³⁰ᴮ⁴⁰ᴰ/UAS-mito::GFP;ato-Gal4¹⁴ᴬ,UAS-CD4::tdTomato/tub-Gal80ᵗˢ*) confirmed the developmental origin of the elongated mitochondria phenotype (**H–I, K**) which is not observed in flies with adult specific depletion (**J, K**). For developmental depletion of *Mirana*, flies were kept at 29 °C until eclosion, and adults after hatching were transferred to 18 °C and kept there for 10–14 days before dissection. For reverse experiment of adult only depletion of *Mirana* (quantifications shown in **K**) flies of

the same genotypes were kept at 18 °C until eclosion, and adults were immediately transferred to 29 °C, and kept there for 10–14 days before dissection. Within developmental stages, Mirana depletion specifically during L3-P48 resulted in elongated mitochondria in adult flies (**L, M, O**). In contrast, *Mirana* depletion at P48-adult (**N, O**) did not. Flies were kept at 18 °C until desired stage of depletion initiation, and then transferred for appropriate time (from L3 till P48, and from P48 till adult) to 29 °C for heat shock inactivation of Gal80 repressor. After heat shock flies were transferred back to 18 °C and were kept there until 14 days old adults before dissection. Mitochondrial size was manually measured through the Z-stack of a confocal image of one/two axons in the optic chiasm and medulla region in 4 different brains of each sex. Scale bar – 10 μm. Normal distribution was checked using the D'Agostino Pearson normality test before statistical comparisons. Non-parametric Kruskal–Wallis test with Dunn's multiple comparison correction was used for mitochondria measurements in fly DCNs neurons. Adjusted *p* values. *$p < 0.05$, **$p \le 0.01$, ***$p \le 0.001$, ****$p \le 0.0001$, ns not significant. Schematics created in BioRender. Hassan, B. (2025) https://BioRender.com/fma66dn.

Taken together, our data show that Mirana is a conserved developmental transcription factor required for axonal mitostasis as well as the establishment of activity-dependent synaptic connectivity and circuit function.

### Mirana regulates mitochondrial morphology and circuit connectivity via developmental regulation of pink1

Mirana regulates several genes involved in mitochondrial dynamics and morphology. To test which target genes are involved in what aspects of the *Mirana* phenotype, we downregulated eight of them independently (*srl/PCG1, CG7772/MTFP1, Drp1/DNM1L, ewg/NRF1, Marf1/MFN2, Atg1/ULK1, parkin/PRKN* and *pink1*) using DCN-specific RNAi.

Knockdown of mitochondrial fission target genes, notably *Mtfp1* and *Drp1*, partially phenocopied the reduction of M-DCN axon numbers (Supplementary Fig. S9), but not mitochondrial elongation. In contrast, knockdown of *Atg1, parkin,* and *pink1*, but not other target genes, resulted in elongated axonal mitochondria (Fig. 6A–D, E). As *pink1* expression in DCNs is quite low at early pupal stages which then gradually increases towards adult stage, we hypothesized that we could mimic elongated mitochondria phenotype by depleting *pink1* levels temporally, after P48, following the Mirana expression peak. Indeed, we observed the same degree of mitochondrial elongation (Fig. 6F–H) as with constitutive *Mirana* or *pink1* knock down, demonstrating a developmental requirement for Pink1.

Next, we asked which target genes were required for circuit connectivity. We found that downregulating *pink1* (but not *Atg1* or *Drp1*) resulted in loss of TransTango labeling, reflecting compromised connectivity with postsynaptic targets (Fig. 6I–K', L). Thus, downregulation of *pink1* phenocopied both mitochondrial morphology and postsynaptic labeling defects, suggesting a mechanistic link between these phenotypes. We therefore asked if restoration of *pink1* expression would at least partially rescue the phenotypes of Mirana loss of function. To this end, we combined *Mirana-RNAi* with *pink1* overexpression in DCNs. This resulted in restoration of both mitochondrial size (Fig. 7A–C') and connectivity to postsynaptic cells back to control levels (Fig. 7F–G'). The initial phenotype we observed upon *Mirana* RNAi was a DCN medulla axon targeting reduction. We did not observe a reduction in the number of medulla axons upon *pink1* downregulation alone. However, to our surprise *pink1* overexpression restored the number of DCN medulla axons to that observed in controls (Fig. 7I–M). We wondered if Mirana depletion reduces mitophagy, thus leading to accumulation of depolarized malfunctional mitochondria. We took advantage of genetically encoded mito-Keima probe[46], to test this option. At the pH of the normal mitochondria matrix (pH 8.0), mito-Keima exhibits an excitation peak at 440 nm, and when delivered to the lysosome, the excitation peak shifts to 586 nm. This large shift allows sensitive detection of the mitophagy process in

vivo[46]. We did not observe a significant difference in mitophagy levels between *Mirana* depleted and control DCNs (Supplementary Fig. S10), supporting the notion that the *pink1*-mediated function of Mirana is required for robust mitochondrial function with no obvious effect on presynaptic mitophagy. Taken together our data suggest that among other putative Mirana targets, mitochondria quality control players *pink1* and *parkin* may be the major regulators of observed effects.

## Discussion

Deciphering the developmental emergence of synaptic connectivity patterns requires understanding the mechanistic relationship between the regulation of gene expression in the neuronal nucleus, and local cell biological processes in distant neurites. In this work we revealed a novel function for a conserved developmental transcription factor we named Mirana which we show acts as a temporal regulator of nuclear genes involved in regulating mitochondrial size required for developmental formation of neuronal circuits (Fig. 8).

The combination of cell type specific transcriptomics and a reverse genetic screen identified several nuclear factors whose developmental down regulation caused connectivity phenotypes. These include transcription factors like *Mirana* and *Sox102F*, and genes such as *Hira*[47–49] and *Brd8*[50], which encode a histone chaperone and a member of the Tip60 chromatin remodeling complex, respectively. Interestingly, several of these show quite striking and complex temporal developmental expression patterns in postmitotic neurons, suggesting that different levels of regulatory control are required at different time points for circuit wiring. Thus, temporal-specific developmental gene regulation in postmitotic neurons represents a general rule underlying neuronal circuit wiring.

In the specific case of TZAP, known for its role in negative regulation of telomere length and mainly studied in the context of cancer biology[23,24], no function was known in the nervous system. Human TZAP/ZBTB48 had been proposed to be a transcriptional activator of a small set of target genes, including *Mitochondrial Fission Process 1* (*MTFP1*)[26] and *Zbtb48* knockout mice show a variety of phenotypes, including abnormal bone structure and eye morphology defects in young adults, and impaired glucose tolerance and behavioral deficits at late adult stages[51], but the underlying mechanisms have not been studied. Our analyses revealed a conserved role for fly Mirana and human TZAP in regulating several key mitochondrial quality control genes relatively early during circuit development. This is then required to ensure normal axonal targeting at early developmental stages, and synaptic activity and functional circuit connectivity at later stages. Given that mitochondrial properties, such as trafficking, change over the course of axon development and neuronal maturation, and that size increase could make it more difficult to affect mitochondria at the extremity of the axon, one explanation for the developmental requirement could be that when Mirana is inhibited in the adult, it is

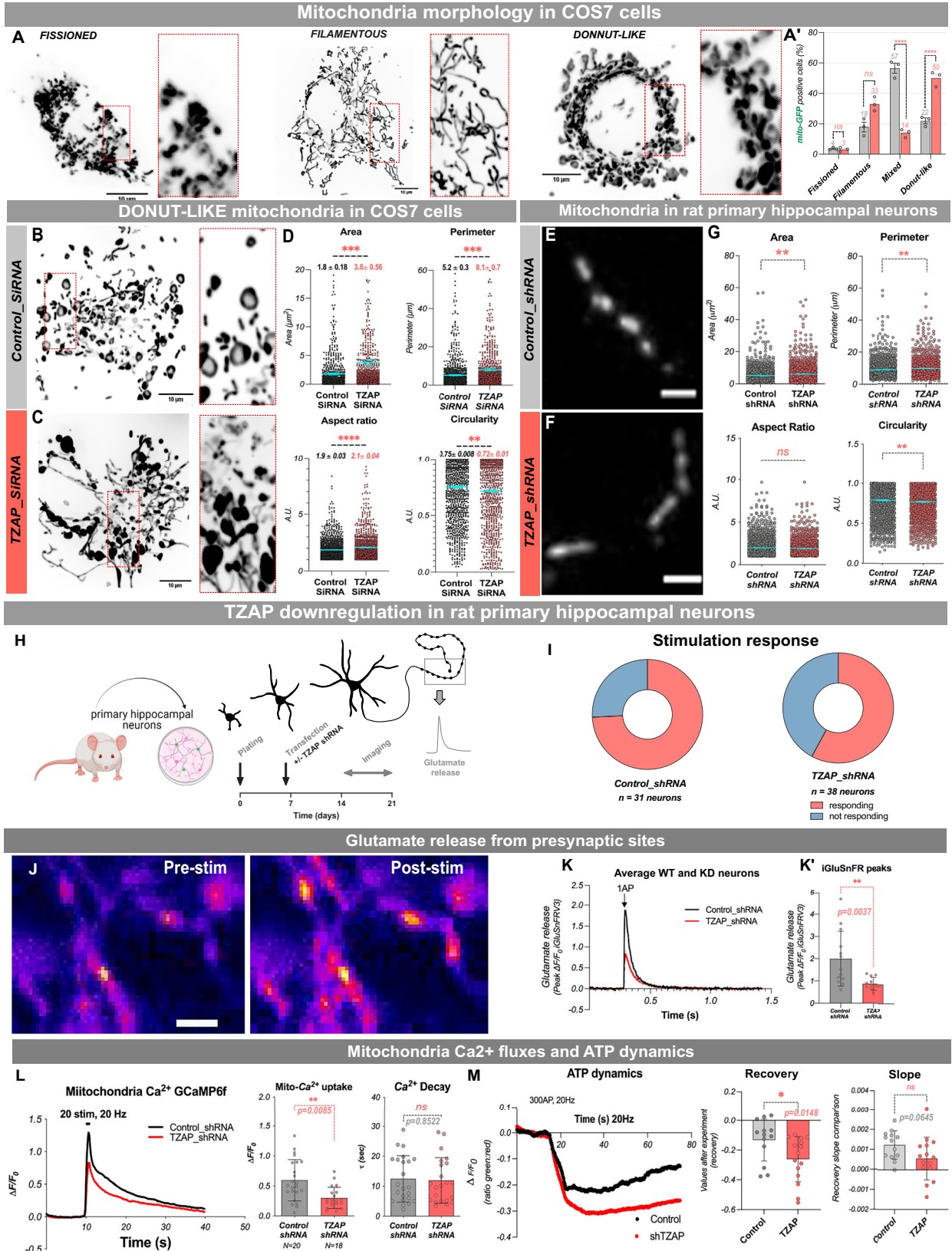

too late to impact the morphology of mitochondria in the axon. Another interesting observation is that overexpression of Mirana did not increase the levels of its target genes. This may be due to either translational silencing of the overexpressed transgene or to a threshold effect, whereby Mirana initiates target gene expression but higher levels do not drive further transcriptional activation. Another

possibility is that Mirana acts as a permissive or pioneer factor with a threshold function. Finally, our data suggest that different aspects of the mitochondrial gene regulatory program are required for different aspects and stages of neuronal circuit development. While mitochondrial fission proteins like Drp1 and dMtfp1 appear to be required during early axon targeting, mitochondrial quality control proteins like

**Fig. 4 | TZAP downregulation in mammalian COS7 cells and primary rat hippocampal neurons leads to defects in mitochondrial morphology and presynaptic neurotransmitter release. A–A'** Mitochondrial morphology of *TZAP* downregulation (*TZAP siRNA*) and control (*Control siRNA*) in mammalian COS7 cells. We quantified the number of cells assigned into four main classes of dominant mitochondrial shapes: fissioned, filamentous, mixed, and donut-like. Downregulation of TZAP leads to a shift in the proportion of mitochondrial morphology towards filamentous and donut-like (**A'**, mean % shown on the graph). Representative images of cells with predominant donut-like mitochondria in control (**B**), and TZAP-RNAi (**C**). Scale bar – 10 μm. **D** Mitochondria morphology parameters (mean ± SEM shown on the graph), $n = 1200$ mitochondria for each condition. **E–G** mitochondria morphology parameters in rat primary hippocampal neurons in control and *TZAP-RNAi* conditions. Scale bar 5 μm, $n = 1397$ control and 1218 *TZAP_siRNA* mitochondria. **H** Scheme of the rat primary neuron experiment created in BioRender. Hassan, B. (2025) https://BioRender.com/fma66dn. Primary neurons derived from rat hippocampus were cultured and sparsely co-transfected with

vectors, carrying iGluSnFR and *control_shRNA* or *TZAP_shRNA* transgenes early at DIV 7 and then assayed upon maturation at DIV 21. **I** Percentage of responding and not-responding cells in both conditions. **J** Representative pseudo color images of glutamate release from presynaptic sites using iGluSnFR sensor in mature neurons (DIV 14-21). **K** Representative average presynaptic responses. **K'** iGluSnFR fluorescence peaks ($\Delta F/F_0$) in both conditions, $n = 13$ control and 12 *TZAP_shRNA* neurons. Scale bar – 5 μm. **L** Representative mitochondria Ca2+ dynamics using GCaMP6f sensor. **M** ATP levels dynamics after stimulation. Data were obtained from 3 biological repeats. Normal distribution was checked using the D'Agostino Pearson normality test before statistical comparisons. A nonparametric two-tailed Mann−Whitney U test was used for mitochondria measurements in rat cortical neurons, two-tailed unpaired t-test to compare ATP recovery slope. COS7 cells data were analyzed by two-way ANOVA, with Tukey post hoc test for multiple comparisons (morphology classification), and unpaired two-tailed t-test (morphology parameters). Exact $p$ values. $P$ value - $^*p < 0.05$, $^{**}p ≤ 0.01$, $^{***}p ≤ 0.001$, $^{****}p ≤ 0.0001$, ns not significant.

Pink1 and Parkin are required for activity dependent circuit connectivity.

Mitochondria play a variety of vital roles. In addition to providing cellular energy, they also contribute to cellular calcium management providing rapid, post-stimulatory $Ca^{2+}$ recovery by taking up massive amounts of calcium with subsequent gradual release into the cytosol[52,53]. These roles are intimately linked with mitochondrial morphology. Previous work showed that elongated mitochondria by MFF KO drove higher mitochondrial Ca2+ uptake upon stimulation[54]. Here, we observe lower uptake suggesting that mitochondrial elongation alone is not the governing factor of mitochondrial Ca2+ changes observed in TZAP KD neurons. Elevated baseline ATP levels in TZAP neurons could indicate an upregulation of ATP-producing pathways that operate independently of activity-driven mitochondrial upregulation, potentially compensating for the apparent decline in activity-dependent mitochondrial ATP synthesis. Further investigation is needed to determine the source of this alternative ATP production.

Our data demonstrate that Mirana loss of function phenotypes can be rescued by restoring the expression of Pink1, suggesting a causal link between mitochondrial morphology and function, and neuronal circuit development. *Pink1* is a key component of the mitochondria quality control system initiating clearance of depolarized mitochondria by mitophagy[55] and regulating mitochondrial $Ca^{2+}$ uptake after stimulation[56]. While we observed $Ca^{2+}$ buffering and ATP production defects, we did not observe strong effects on mitophagy upon *Mirana* knockdown. This is consistent with the emerging evidence for mitophagy-independent functions of Pink1 and Pink1-independent basal mitophagy in vivo[57,58]. Our data suggest that even when the level of expression of other genes is reduced, presence of excess amounts of *Pink1* may be enough to ensure a level of mitochondrial function compatible with normal circuit development. Further work is needed to elucidate the downstream signaling modules and cellular processes regulated by Mirana and Pink1 in the context of mitochondrial dynamics and synaptic connectivity.

Strikingly, both Mirana and Pink1, whose loss in humans causes Parkinson's disease, appear to be specifically required during development for mitochondrial size regulation and DCN circuit wiring. Even when their expression was restored for several days or even weeks after a relatively brief developmental knockdown, the phenotypes remained. Considering the increasing evidence for a developmental origin of neurodegenerative diseases[59], it will be important to determine the contribution of the developmental function of neurodegenerative disease genes like Pink1 and the ataxia-associated mitochondrial fission gene *DNM1L/Drp1*[60], to adult neuronal dysfunction. Insights gained from future studies of the role of developmental regulation of mitochondrial biology in human neuronal circuit development could inform the development of novel biomarkers and therapeutic targets for neurological disorders characterized by synaptic dysfunction.

## Methods
### Experimental model and subject details
*Drosophila melanogaster* were raised on a standard cornmeal/agar diet (8 g Agar, 60 g cornmeal, 50 g yeast, 20 g glucose, 50 g molasses, 19 ml ethanol, 1.9 g Nipagin and 10 ml propionic acid in 1 L of water). Animals were raised in groups up to 20 until 5 days old at 25 °C in a 12/12-h light/dark regime at 60% humidity.

For developmental depletion of *Mirana*, Control and *Mirana*-RNAi carrying flies were kept at 29 °C until eclosion, and adults after hatching were transferred to 18 °C and kept there for 10–14 days. For the reverse experiment of adult only depletion of *Mirana*, Control and *Mirana*-RNAi carrying flies were kept at 18 °C until eclosion, and adults were immediately transferred to 29 °C, and kept there for 10–14 days.

For temporal temperature-controlled depletion (Figs. 2, 3, and 6), Control and *Mirana*-RNAi carrying flies and crosses were kept at 18 °C until L3 stage, and then moved to 29 °C for desired time of depletion. After this period pupae were moved back to 18 °C and kept at this temperature until dissection as 7–10 days old adults.

For the behavioral experiment on day 5, the wings were cut under $CO_2$ anesthesia. They were left to recover for 48 h within individual containers with access to fresh food before being transferred to the experimental set-up.

In fly experiments we examined progeny of correct genotype from at least three independent crosses, and dissected around 10 flies of each gender for imaging and further analysis. This procedure was mainly done once for each genotype in RNAi screen, and at least twice for the rest of experiments.

### Cryosections of adult fly heads
Cryosections were prepared for further Laser Micro Dissection of DCN clusters. Fly heads were cut from the body and aligned in plastic molds (on ice) filled with Tissue-Tek O.C.T. Compound and immediately frozen on dry ice. Before sectioning blocks were kept at −20 °C in a cryostat chamber for 10 min to adjust their temperature and avoid cracking during trimming. 15 μm frozen sections were performed using Leica CM3050S Cryostat. Sections were put on PEN-Membrane Slides (Leica, #11505189) and kept at −80 °C in boxes containing Silica gel beads until the LMD procedure.

### Laser micro dissection
Was performed from the dehydrated brain cryosections following described protocol[18,61] followed by total RNA isolation. DCN clusters were visualized on cryosections under UV light by two fluorescence markers (CD8::GFP in membranes, and RedStinger in cell nucleus). Control samples were formed from random cuts of central brain regions from the same brains. Dissected tissue was collected (-25 clusters (from numerous animals)/sample) into the RNAse free PCR-tubes containing 0.2% Triton X100 in DEPC treated water (Invitrogen

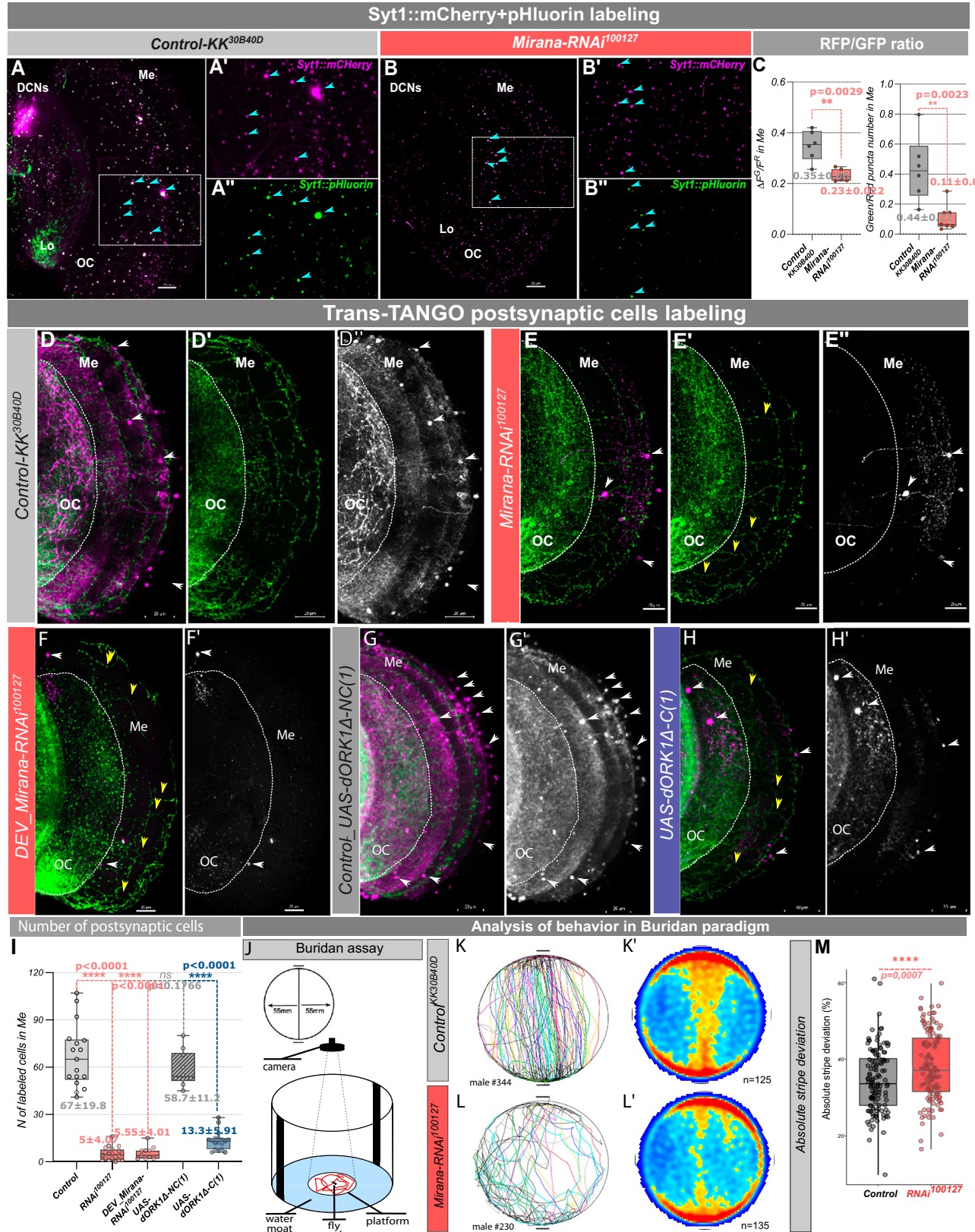

**Syt1::mCherry+pHluorin labeling**

*Control-KK³⁰ᴮ⁴⁰ᴰ* — *Mirana-RNAi¹⁰⁰¹²⁷* — RFP/GFP ratio

**Trans-TANGO postsynaptic cells labeling**

Number of postsynaptic cells — Analysis of behavior in Buridan paradigm

#AM9916) with 1 µl of RNases inhibitor solution (RNaseOUT Recombinant Ribonuclease Inhibitor, Quiagen #10777019). Samples were stored at −80 °C until RNA isolation.

Total RNA was isolated using the Ambion RNaqueous Micro Kit adapted for LMD samples (#1931). RNA concentration was checked using TapeStation, High Sensitivity D1000 Screen Tape (Agilent).

**RNA sequencing**

mRNA library preparation was realized following the manufacturer's recommendations using SMART-Seq v4 Ultra Low Input RNA Kit from TAKARA with Nextera Tagmentation Kit. Final samples pooled library preps were sequenced on NGS NextSeq 500 Illumina using a High Output cartridge (2x400Millions of 150 bases reads).

**Fig. 5 | Mirana downregulation reduces synaptic vesicle release and connectivity to postsynaptic cells. A–B″** Live imaging of steady state synaptic vesicle release using *UAS-Syt1-mCherry::pHluorin* pH sensitive fluorescence probe in control (**A–A″**) and *Mirana-RNAi^{100127}* (**B–B″**). **C** Ratio of green (pHluorin, cyan arrowheads) to red (mCherry) fluorescence and number of puncta in control (n = 6) and *Mirana-RNAi* condition (n = 7). **D–F′, I** TransTango anterograde postsynaptic labeling of *Mirana* downregulation, with an expression of *UAS-Mirana-RNAi^{100127}* (n = 20 optic lobes) *and Mirana_gRNA; UAS-U^M-Cas9* (Supplementary Fig. S5, n = 25 optic lobes) compared to control (n = 17 optic lobes). Visible M-DCN axons in yellow arrowheads. Developmental *Mirana* depletion (from L1 till eclosion, **F–F′, I**, n = 11 optic lobes). **G, G′** Silencing of DCNs by overexpression of conductive form of potassium channel dORK (*UAS-dORK1Δ-C(1)*, n = 18 optic lobes). **H, H′** Non-conductive form of dORK (*UAS-dORK1Δ-NC(1)*, n = 10 optic lobes) used as an additional control, showed no effect on TransTango labeling. DCNs expressing myrGFP on the cellular membrane are shown in green (**D, D′, E, E′, F, G, H**) and postsynaptic cells in magenta (merged picture) and grayscale (**D, D″, E, E″, F, F′, G, G′, H, H′**).

Postsynaptic connectivity was analyzed by manually counting cell bodies in Medulla (white arrowheads) labeled with dsRed fluorescence (magenta, merged picture – **D–H** and grayscale – **D″, E″, F′, G′, H′**), including all cells with weak or strong labeling to reveal all potential connections. Scale bar 20 μm. Object orientation behavior of flies with *Mirana* downregulation in DCNs in Buridan assay (**J**). **K, L** Individual fly trajectory during recording. **K′, L′** Heat map of arena occupancy at the population level, n = 125 control and 135 *RNAi^{100127}* flies. **M** Absolute stripe deviation measurement, showing that Control flies (*W+; UAS-KK^{40D30B};ato-Gal4^{14A},UAS-CD4::GFP*) normally walk more straight (represented in lover ASD value) than flies with *Mirana* downregulation (*W+;UAS-Mirana-RNAi;ato-Gal4^{14A},UAS-CD4::GFP*). Me medulla, Lo lobula, OC optic chiasm. Ordinary one-way ANOVA with the Kruskal–Wallis test was used for the behavioral experiment and unpaired t-test with Welch's correction for synaptic marker counting and trans-TANGO labeling analysis. Exact p values. *p < 0.05, **p ≤ 0.01, ***p ≤ 0.001, ****p ≤ 0.0001, ns not significant.

Corresponding to 2x25Millions of reads per sample after demultiplexing.

## RNA sequencing data analysis

Reads were aligned to the *Drosophila melanogaster* genome (r6.10) using STAR (2.5.4)[62]. We identified differentially expressed genes in adult clusters (Fig. 1) by performing a Mann–Whitney U test on each DCN sample compared to the random central brain cuts samples. We used a multidimensional scaling analysis to exclude poorly clustering outlier samples. *p*-Values were adjusted using Benjamini-Hochberg procedure (Supplementary Table S1). Putative *ato*-targets were found using i-cisTarget software. Genes with an adjusted *p*-value < 0.01, base pair reads >100 bp, and 2LogFold Change >2 were used in the follow-up analysis. The upregulated gene list was manually curated by gene function annotation obtained from FlyBase and used for RNAi-screen (Supplementary Fig. S1). RNA-seq data analysis was performed with R and RStudio (DESeq2 package). Sequencing reads and pre-processed sequencing data are available in the NCBI Gene Expression Omnibus.

## qRT-PCR (*Drosophila* brains)

Total RNA for qRT-PCR analysis was isolated using column-based RNeasy Micro Kit (Qiagen) from dissected *Drosophila* L3 brains (10–15 brains/sample), and beads-based RNAdvance Cells v2 Kit (Beckman Coulter) from adult fly heads (10–15 heads/sample) following the manufacturer's protocol. Reverse transcription was performed starting with 500 ng/μl total RNA concentration, by using one step Verso cDNA Synthesis Kit (Thermo Fisher Scientific). qPCR was performed on QPCR LC480 cycler (Roche), using Light Cycler 480 SYBR Green I Master Mix (Roche), and self-designed and tested primers (Key reagents table). The experiments were performed with at least three independent biological samples and carried out with technical triplicates. Data are shown as fold change, calculated as $2^{-\Delta\Delta Ct}$ using *Act88F* as a calibrator gene.

## ChIP-seq analysis

ChIP-seq analysis was done using the Galaxy Project (https://usegalaxy.org/). Reads were trimmed using Cutadapt (--max-n 4) and Trimmomatic (TRAILING 1; SLIDINGWINDOW 4 and cut off 20; LEADING 20; MINLEN 50) and mapped using Bowtie2 onto mm10 mouse reference genome (-X 600; -k 2; --sensitive). PCR-derived duplicates were removed using PICARD MarkDuplicates. Bigwig files were generated with bamCoverage (bin size = 1). Peak calling was performed using MACS2 callpeak with Input as control and with options: --qvalue 0.05; --nomodel; --keep-dup 1. Regions found in KO were then removed using bedtools Intersect intervals. Visualization of coverage and peaks was done using IGV[63]. Analysis was done for HeLa and U2OS cell lines, with corresponding KO. Peaks found from peak calling are shown as rectangles below coverage. Intersections between WT and KO were

done using bedtools Intersect intervals to keep the peaks only present in WT samples. For fly data, coverage and peak mapping were available[25]. Visualization of coverage and peaks was done using IGV[63].

## Immunofluorescence of *Drosophila* brains

For brain immunofluorescence labeling L3 instant larvae brains and 3–7 days old fly brains were dissected in 1X-PBS and transferred to a tube for fixation in 4% paraformaldehyde in PBS for 20 min at room temperature. Brains were washed in PBS (one time) and 0.3% Triton-X (PBST) three times for 10 min each. Next, brains were placed in blocking solution (2% normal bovine serum (Sigma-Aldrich) in PBST) overnight at 4 °C. The samples were then incubated in primary antibodies diluted in working solution (1 ml of 2% BSA, 250 μl 10% NatAzid in 50 ml 2% PBST) overnight at 4 °C. Used primary and secondary antibodies are listed in the Key Reagents table. Brains were then washed three times in PBST for 10 min each and incubated with a fluorochrome-conjugated secondary antibody overnight at 4 °C (listed in the Key Reagents table). The brains were washed three times in PBST for 10 min each. Finally, the samples were mounted onto microscope slides with Vectashield Vibrance Antifade Mounting Medium (Vector Laboratories) for subsequent analysis using the SP8 Leica or Olympus FV1200 confocal microscopes. Images were analyzed using the Fiji ImageJ and IMARIS software.

Trans-TANGO experiment was performed with the DCN-specific *ato-Gal4*^{14a} driver line, and the number of postsynaptic neurons was counted manually from their cell bodies, including all cells with weak or strong labeling to reveal all potential connections.

pHluorin to mCherry fluorescence ratio detection was made in freshly dissected brains in PBS on ice. After dissection, brains were immediately mounted on the Superfrost slides with Vectashield Antifade Mounting Medium (Vector Laboratories) and imaged on Confocal SP8 Leica. Mito-GFP images for dendritic mitochondria analysis were taken on the Spinning Disk 3i/Zeiss|Upright ICM.Quant (RRID:SCR_026393). Impact on mitochondria morphology was assessed using ImageJ software (NIH, Bethesda, Maryland) using the diameter of mitoGFP-positive objects.

## Behavioral arena

The behavioral arena used is a modification of Buridan's Paradigm[44]. The arena consists of a round platform of 119 mm in diameter, surrounded by a water-filled moat (Fig. 6H). The arena was placed into a uniformly illuminated white cylinder. The setup was illuminated with four circular fluorescent tubes (Philips, L 40w, 640C circular cool white) powered by an Osram Quicktronic QT-M 1 × 26–42. The four fluorescent tubes were located outside of a cylindrical diffuser (Canson, Translucent paper 180gr/m²) positioned 145 mm from the arena center. The temperature on the platform during the experiment was maintained at 25 °C.

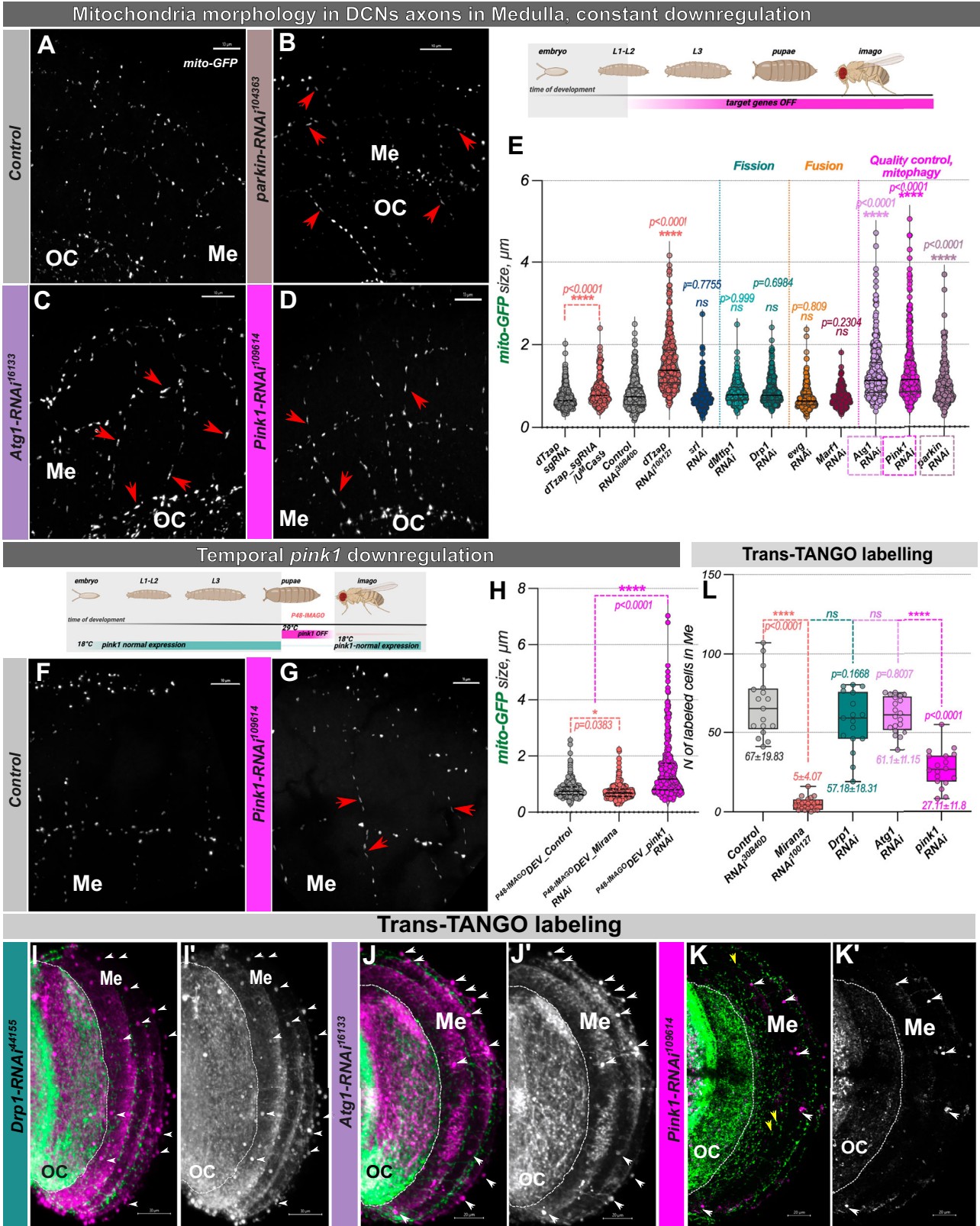

### ChIPseq available data analysis
Available ChIPseq data from whole-organism adult *Drosophila* females were available at https://www.encodeproject.org/experiments/ENCSR472BOK/[25]. Human ChIPseq data were published by Jahn et al.[42]. Drosophila reads were aligned with the dm6 reference *Drosophila* genome, human – with hg38 reference

genome using Integrative Genome Viewer IGV and Genomatix software.

### Statistical analysis
Data from behavioral experiments were analyzed using R[64], and from other experiments – using Prism-GraphPad software. In behavior

**Fig. 6 | Mirana regulates the morphology of mitochondria in flies through expression tuning of downstream genes. A–E** Mitochondria (labeled with mito-GFP) in DCNs axons under constitutive downregulation of Mirana target genes. **F–H** Temporal *pink1* downregulation during development (*pink1-RNAi^{i09614}/UAS-mito::GFP;ato-Gal4^{144},UAS-CD4::tdTomato/tub-Gal80^{ts}* vs *UAS-KK^{30B40D}/UAS-mito::GFP;ato-Gal4^{144},UAS-CD4::tdTomato/tub-Gal80^{ts}*) during P48-adult time window (time point following Mirana peak expression) recapitulates the elongated mitochondria phenotype. For developmental depletion of *pink1*, flies were kept at 18 °C until P48, then transferred to 29 °C until eclosion, and adults after hatching were transferred to 18 °C, and kept there for 10–14 days before dissection. Mitochondria size was manually measured through the Z-stack of a confocal image of one/two axons in the Optic chiasm and Medulla region in at least 3/4 different brains of each sex. Scale bar – 10 µm. **I–L** TransTango labeling of postsynaptic

connectivity in *Drp1* (*Drp1-RNAi^{44155}*, *n* = 17 optic lobes), *Atg1* (*Atg1-RNAi^{16133}*, *n* = 20 optic lobes), and *pink1* (*pink1-RNAi^{i09614}*, *n* = 18 optic lobes) downregulation in DCNs. **L** Number of postsynaptic cells was calculated by manually counting cell bodies in the medulla (white arrowheads) labeled with dsRed fluorescence (magenta and grayscale), including all cells with weak or strong labeling to reveal all potential connections. In case of *pink1* downregulation, just like in *Mirana* silencing condition, we can find M-DCNs axons in Medulla (yellow arrowheads), however, TransTango labelling is severely decreased (**K, K'**, *n* = 18 optic lobes). Scale bar – 20 µm. Statistical analysis was done using the nonparametric Kruskal–Wallis test with Dunn correction for multiple comparisons and one-way ANOVA with Bonferroni correction for multiple comparisons respectively. Adjusted *p* values. *$p < 0.05$, **$p \leq 0.01$, ***$p \leq 0.001$, ****$p \leq 0.0001$, ns not significant. Schematics created in BioRender. Hassan, B. (2025) https://BioRender.com/fma66dn.

analysis transition plots were done as described before[44]. Briefly, the platform was divided into 60*60 hexagons and the fly's position raised the count of each hexagon by one in the arena. The scale starts at 0 (blue) and goes up till a value is calculated by the 95%-quantile of the count distribution (red). Normal data distribution was first tested using the Shapiro–Wilk normality test. non-parametric tests were further performed: a two-sided Mann–Whitney U-test or a rank-based non-parametric Kruskal–Wallis test followed by a Dunn test when comparing more than two conditions. In behavioral data analysis, we used Ordinary one-way ANOVA with the Kruskal–Wallis test. In other experiments, we used Ordinary one-way ANOVA with Dunnett and Bonferroni corrections for multiple comparisons and unpaired Student's *t* test with Welch's correction. For axon development experiment we estimated mean difference between the different groups, using dabestr package [v2025.3.14] from R using default settings, with a confidence interval of 95%. The bootstrap method was used as a resampling method (5000 bootstrap samples) to build confidence interval without relying on strict assumptions. The confidence interval is bias-corrected and accelerated. *$p < 0.05$, **$p \leq 0.01$, ***$p \leq 0.001$, ****$p \leq 0.0001$, ns not significant.

## Studies in mammalian cells

**Cell culture and transfection of COS7 cells.** COS7 cells (ATCC) were grown in Dulbecco's modified Eagle medium (DMEM, Gibco) supplemented with 10% fetal bovine serum (FBS), 1% L-glutamine (Invitrogen), and 1% penicillin-streptomycin (Invitrogen) and tested monthly for mycoplasma contamination (MycoAlert Mycoplasma Detection Kit, Lonza). Cells were plated in 8-well glass bottom Ibidi (80827; Ibidi), at a density of $1.5 \times 10^4$ per well precoated with poly-L-lysine (1 mg/L, Sigma). The following day, cells were transfected with 0.1 µg/well of pcDNA3-HA, 0.1 µg/well of pCB6-HA-mitoGFP[65], and 10 pmol/well of TZAP stealth siRNA (s6568, ThermoFisher) or AllStars negative control (Qiagen 1027281), with Lipofectamine 2000 (0.5 µl/well, Invitrogen) in 50 µl of Opti-MEM I (Gibco), without medium change, according to the manufacturer's instructions.

**Microscopy and image analysis of COS7 cells.** After 48 hours, Z-stack projection images were captured in life-imaging with a Spinning disk CSU-X1 confocal microscope (Leica) driven by MetaMorph software (63X immersion oil objective, N.A. 1.4), and processed with ImageJ software. Cells expressing MitoGFP in images were manually categorized according to mitochondrial network morphology, classified as fissioned; donut-like; fusioned and mixed (with no predominant type of morphology). The results are expressed as mean values ± SEM from three independent experiments.

For the quantitative analysis of the morphology of the mitoGFP-positive mitochondrial objects, a manual threshold mask was applied on each of 25–30 images per condition, using ImageJ software (NIH, Bethesda, Maryland). For each mitochondrial object, area, perimeter, aspect ratio, and form factor were calculated. Aspect ratio is the ratio between the major and minor axes of each mitochondrial object, and

represents its length; form factor is calculated as perimeter2/(4π × area) and thus represents a combined evaluation of the length and degree of branching of the mitochondrial network. We analyzed a total of 940–1200 objects per condition from three independent experiments. The results are expressed as means ± SEM.

**Quantitative analysis of transcript levels.** To evaluate the efficiency of the siRNA-mediated silencing approach for endogenous TZAP and expression levels of genes involved in the regulation of mitochondrial dynamics, total RNA was extracted from COS7 cells, using RNeasy Micro kit (Qiagen), according to the manufacturer's instructions, and treated with DNAse I (Qiagen) for 20 min at room temperature. RNA concentrations were determined using Nanodrop 2000c (THERMO Scientific). Complementary DNA was generated from 500 ng RNA with random hexamers and Superscript II reverse transcriptase (THERMO Fisher Scientific). Real-time PCR was performed with the LightCycler 480 qPCR system (Roche) with SYBR green detection and each of the following primer pairs in 5'-to 3' orientation (see Key reagents table). Relative gene expression levels were calculated using the $2^{-\Delta\Delta Ct}$ method, with GAPDH used as the reference gene for normalization. Statistical analyses were performed using GraphPad Prism 7 software. Normal distribution was checked using the D'Agostino Pearson normality test before statistical comparisons. Data were analyzed by two-way ANOVA. When pertinent, the Tukey post hoc test was used for multiple comparisons. The sample size was standard and determined based on previous experience. Analyses were not blinded and no sample was excluded. The results are presented as means ± SEM from three independent experiments.

## Studies in rat primary hippocampal neurons

**Animals.** Wild-type rats were of the Sprague-Dawley strain Crl: CD (SD) and were bred by Janvier Labs (France) following the international genetic standard protocol (IGS). All procedures relating to the care and treatment of animals were performed following the guidelines of the European Directive 2010/63/EU and the French Decree n° 2013-118 concerning the protection of animals used for scientific purposes.

**Primary co-culture of postnatal hippocampal neurons and astrocytes.** Imaging experiments for mitochondrial morphology and glutamate release were performed in primary co-cultures of hippocampal neurons and astrocytes using P0 to P2 rats of mixed gender. After dissection to isolate the hippocampus, cells were plated on cloning cylinders of 4.7 mm diameter attached to poly-ornithine-coated coverslips (38,000–40,000 cells per cylinder) and maintained in culture media composed of MEM (Thermo Fisher Scientific, #51200087), 20 mM Glucose (Sigma, #G8270), 0.1 mg/mL transferrin (Sigma, #616420), 1% GlutaMAX (Thermo Fisher Scientific, #35050061), 24 µg/mL insulin (Sigma, #I6634), 10% FBS (Thermo Fisher Scientific, #10082147) and 2% N-21 (Bio-techne, #AR008). After 4–5 days in vitro, neurons were fed with media of the same composition but with only 5%

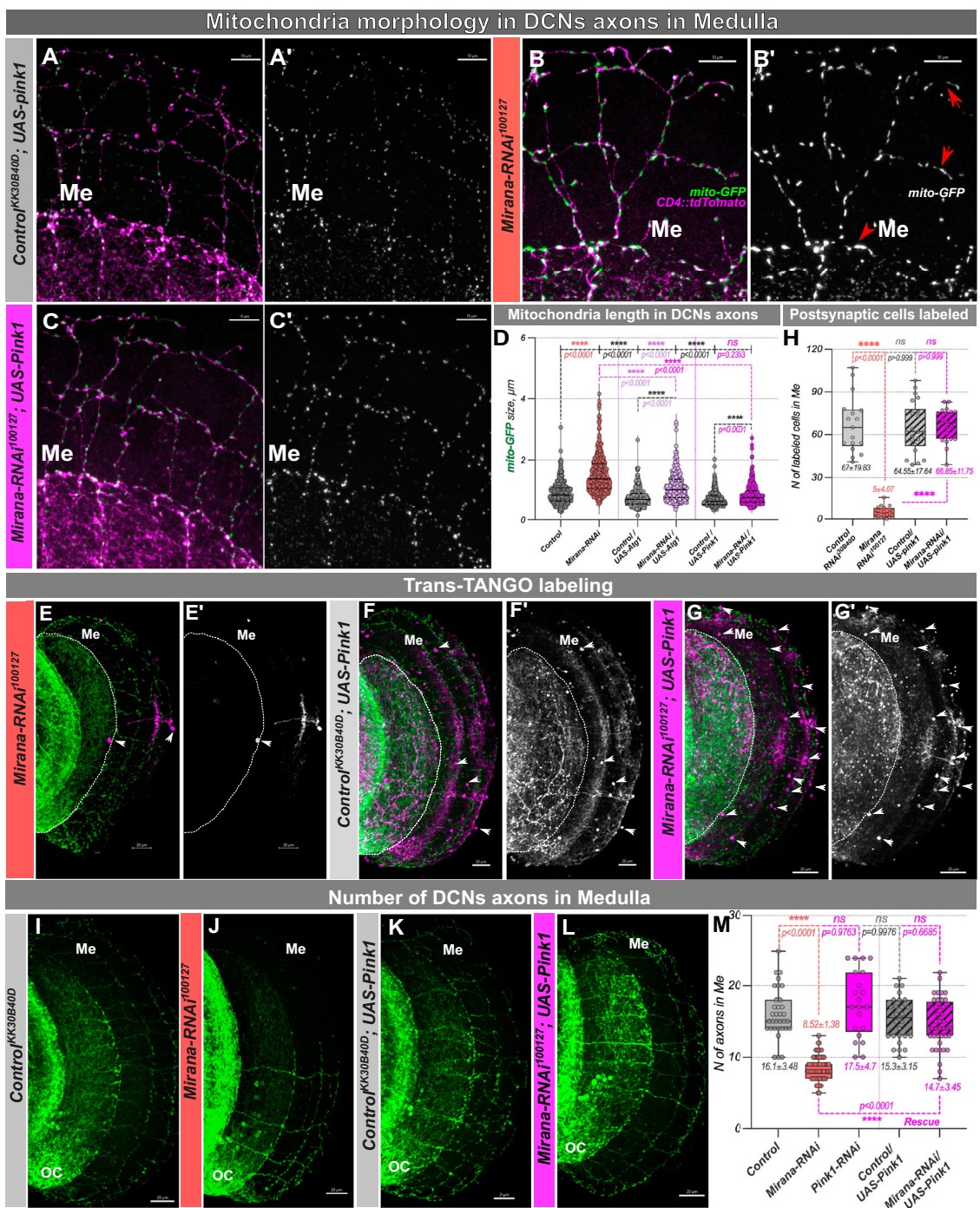

**Fig. 7 | Overexpression of pink1 in DCNs rescues all Mirana loss of function phenotypes. A–C'** Mitochondria size (labeled with mito-GFP) in DCNs axons was rescued to that of controls (w[1118];*UAS-mito::GFP/UAS-KK[40D30B]*,*ato-Gal4[14A]*,*UAS-CD4::tdTomato/UAS-pink1*, **A–A'**) with *pink1* overexpression (w[1118];*UAS-mito::GFP/UAS-Mirana-RNAi[100127]*,*ato-Gal4[14A]*,*UAS-CD4::tdTomato/UAS-pink1*, **C–C'**). Mitochondria size was manually measured through the Z-stack of a confocal image of one/two axons in the optic chiasm and medulla in at least 3/4 different brains of each sex (**D**). Scale bar – 10 μm. **H–G'** Anterograde transsynaptic labeling of postsynaptic partner cells using the TransTango approach (**E'**, **G'**, n = 22 optic lobes for w[1118];*UAS-mito::GFP/UAS-KK[40D30B]*,*ato-Gal4[14A]*,*UAS-CD4::tdTomato/UAS-pink1* and 20 optic lobes for w[1118];*UAS-mito::GFP/ UAS-Mirana-RNAi[100127]*,*ato-Gal4[14A]*,*UAS-CD4::tdTomato/UAS-pink1*). Postsynaptic connectivity was calculated by manually counting cell bodies in Medulla (white arrowheads) labeled with dsRed fluorescence (magenta and grayscale) (**H**, n = 17 optic lobes for *Control-RNAi[30B40D]*. N = 20 optic

lobes for *Mirana-RNAi[100127]*. N = 22 optic lobes for *Control-RNAi[30B40D]*,*UAS-pink1*. N = 20 optic lobes for *Mirana-RNAi[100127]*,*UAS-pink1*), including all cells with weak or strong labeling to reveal all potential connections. Data are presented as mean values +/− SEM. Scale bar – 20 μm. **I–M** Number of DCN axons reaching the Medulla in adult flies. *pink1* overexpression (*ato-Gal4[14A] > UAS-Mirana-RNAi[100127]*,*UAS-pink1*) restored the M-DCNs axons, caused by *Mirana* downregulation (**M**, n = 33 optic lobes for *Control*. N = 16 optic lobes for *Mirana-CRISPR/Cas9*, n = 21 optic lobes for *pink1-RNAi* n = 26 optic lobes for *Control,UAS-pink1*, n = 40 optic lobes for *Mirana-RNAi[100127]*,*UAS-pink1*. Measurements were done equally in flies of both sexes. Data are presented as mean values ± SEM. Me medulla. Scale bar − 20 μm. Statistical analysis was done using one-way ANOVA with Dunnett, and Bonferroni corrections for multiple comparisons in mitochondria length comparison. Adjusted p values. *p < 0.05, **p ≤ 0.01, ***p ≤ 0.001, ****p ≤ 0.0001, ns not significant.

Fig. 8 | Model of the temporal regulation of visual circuit development and the role of Mirana in the process. mRNA of *Mirana* is detected at early pupal development, with peaking at P15–P30 APF, followed by Mirana protein peak expression levels at P30–P40 APF. This timepoint of temporal expression is a critical time window during which Mirana regulates expression of its target genes, including *Pink1*, and primes mitochondria health to ensure proper axon growth and circuit connectivity leading to appropriate visual responses. Created in BioRender. Hassan, B. (2025) https://BioRender.com/fma66dn.

FBS and adding 2 μM cytosine β-d-arabinofuranoside to stop glial growth (Sigma, #C6645). Neurons were transfected 6-7 days after plating using calcium phosphate as previously described[66]. The co-transfection mix (DNAs-Ca$^{2+}$-PO$_4$ precipitate) was prepared using 0.63 μg of each DNA per dish. Cultures were maintained at 37 °C in a 95% air/5% CO$_2$ humidified incubator prior to carry out imaging experiments, which were performed from DIV14 to DIV21.

**Plasmid constructs for live cell imaging in rodent neurons.** The following DNA constructs were used for expressing biosensors for live cell imaging: mito$^{4x}$-GCaMP6f[32] and iGluSnFR3[67], which was a gift of Kaspar Podorgski (Addgene plasmid #178330) and iATPSnFr2-Halo[37], which was a gift from Timothy A. Ryan. The shRNA sequence for the Zbtb48 gene (ENSRNOG00000009595.6) was generated from the GPP Web Portal (https://portals.broadinstitute.org/gpp/public/seq/search). The selected shRNA sequence 5'-AGGGATGGTGATGGCGAT-TAT-3' was cloned into pLKO.1-TRCmTagBFP2, a gift from Timothy A. Ryan (Addgene plasmid #191566), using the AgeI and EcoRI sites as previously described[68]. The primers to generate the insert containing the shRNA sequence are annotated in the Key Resources table. For control experiments we used the same pLKO-mTagBFP2 plasmid without inserting an shRNA.

**Live imaging of neurons.** Neuronal imaging was performed in continuously flowing Tyrode's buffer containing (in mM) 119 NaCl, 2.5 KCl, 1.2 CaCl$_2$, 2.8 MgCl$_2$, 20 glucose, 10 μM 6-cyano-7-nitroquinoxaline-2,3-dione (CNQX) and 50 μM D,L-2-amino-5-phospho-novaleric acid (AP5), buffered to pH 7.4 at 37 °C using 25 mM HEPES. Imaging was performed on a custom-built laser-illuminated epifluorescence microscope (Zeiss Axio Observer 3) coupled to an Andor iXon Ultra camera (model #DU-897U-CSO-#BV), whose chip temperature is cooled down

to −90 °C using the Oasis™ UC160 Cooling System allowing the reduction of noise in the measurements. Illumination using a fiber laser of wavelength 488 (Coherent OBIS 488 nm LX 30 mW) was controlled by a custom Arduino-based circuit coupling imaging and illumination.

Primary hippocampal neurons were cultured on 0.17 mm coverslips (Warner Instruments, #640705), mounted in an imaging chamber designed for field stimulation (Warner Instruments, #RC-21BRFS), and imaged using a Zeiss 40x oil-immersion objective (Plan-Neofluar, NA 1.30, WD = 0.21 mm). For glutamate release measurements using iGluSnFR3, imaging was performed at 350 Hz during single action potentials elicited by 1 ms field stimulation. Mitochondrial Ca$^{2+}$ dynamics were quantified imaging mito$^{4x}$-GCaMP6f responses at 5 Hz. Given that mito$^{4x}$-GCaMP6f occasionally mislocalizes to the cytosol, mitochondrial Ca$^{2+}$ peak responses were extracted after cytosolic Ca$^{2+}$ clearance to ensure accurate detection, as previously described[32].

Dual imaging experiments for monitoring intracellular synaptic ATP levels were performed using a custom-built laser illuminated epifluorescence microscope (Zeiss Axio Observer 3), whose chip temperature is cooled down to −70 °C, coupled to two Andor iXon Life cameras (model IXON-L-897) through of two-camera imaging adapter (Andor TuCam-High Performance). During image acquisition, laser excitation wavelengths were alternated between frames using OBIS 488 nm LX and OBIS 561 nm LS lasers, which were combined via the Coherent Galaxy Beam Combiner. Illumination and imaging were synchronized through a custom Arduino-based circuit, as in previous experiments. Prior to perform the experiment, primary hippocampal neurons transfected with synaptic-iATPSnFR2-Halo construct were incubated for 30 min with Janelia Fluor 585 HaloTag Ligand (Promega, #HT104A) at 500 nM final concentration in cell media to label the HaloTag, and washed three times with Tyrode's buffer before to be mounted on the imaging chamber for field stimulation. To obtain

activity-driven intracellular ATP changes, neurons were stimulated with 300 AP fired at 20 Hz and images were acquired at 2 Hz.

The temperature of all experiments was clamped at 37 °C and was kept constant by heating the stimulation chamber through a heated platform (Warner instruments, #PH-2) together with the use of an in-line solution heater (Warner instruments, #SHM-6), through which solutions flowed at 0.35 ml/min. The temperature was monitored constantly using a feedback-look temperature controller (Warner instruments, #TC-344C).

**Image analysis, statistics, and exclusion criteria.** Image analysis was performed using the ImageJ plugin Time Series Analyzer v3, selecting ~15–25 ROIs per neuron for mitochondrial $Ca^{2+}$ (including non-responding mitochondria) and intracellular ATP measurements, or ~3–4 ROIs per neuron for glutamate release. Fluorescence intensity was measured over time. Glutamate release and mitochondrial $Ca^{2+}$ in response to electrical activity ($\Delta F$) were normalized to the resting fluorescence ($F_0$). A criteria of inclusion/exclusion data was established to avoid overestimating $\Delta F/F_0$ in responding neurons with low $F_0$ values, setting an arbitrary threshold such that $F_0$/background >1.25 to be included for further analysis (11/49 Mito$^{4x}$-GCaMP6f and 0/25 iGluSnFR3 were excluded).

ATP signals were quantified as the ratio of iATPSnFR2 to HaloTag fluorescence (Green:Red). Imaging was performed by alternating illumination with 488 nm and 561 nm lasers, and emissions from each fluorophore were directed to separate cameras to prevent bleed-through. Synaptic bouton ROIs were identified based on Halo dye fluorescence. The Green:Red ratio change ($\Delta R$) in response to electrical activity was normalized to the baseline Green:Red ratio fluorescence ($R_0$).

Statistical analysis was performed with GraphPad Prism v8.0 for Windows using the nonparametric Mann-Whitney U test to determine the significance of the difference between two unpaired conditions, as we did not assume that the distributions of our two unpaired datasets follow a normal distribution. $p < 0.05$ was considered significant and denoted with a single asterisk, whereas $p < 0.01$, $p < 0.001$, and $p < 0.0001$ are denoted with two, three, and four asterisks, respectively. The $n$ value represents the number of cells imaged and statistic tests performed in each experiment are indicated in the respective figure legends. Experiments were replicated with neurons from at least 4 independent primary cultures per condition.

**Mitochondrial shape parameters.** Different morphological aspects of mitochondria were analyzed from the images taken in the mitochondrial calcium measurement experiments. Images were analyzed using the ImageJ plugin Trainable Weka Segmentation following the criteria described previously[69]. For cultured neurons, in addition to the global background of the image not containing any cell type, pixels containing astrocytes were also selected as a secondary background, which increased specificity in the mitochondrial shaping measurements performed by the Trainable Weka Segmentation plugin. The morphological aspects of mitochondria analyzed for neurons were area, perimeter, circularity, and aspect ratio, the values were obtained in pixels and later converted to micrometers following the characteristics of the camera used (1 pixel = 0.4 μm).

**Reporting summary**
Further information on research design is available in the Nature Portfolio Reporting Summary linked to this article.

## Data availability
All data supporting the findings of this study are available within the paper and its Supplementary Information. Source data are provided with this paper.

## Code availability
No new code was generated as part of this work.

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

## Acknowledgements

This work was supported by the Investissements d'Avenir program (ANR-10-IAIHU-06), Paris Brain Institute-ICM core funding, The Allen Distinguished Investigator Award (ADI12202), Roger De Spoelberch Prize, NIH Brain Initiative RO1 grant (1R01NS121874-01), and ANR grant QuantSocInd (ANR-19-CE16-000) (to B.A.H.). We gratefully acknowledge C. Lovo from the ICM.Quant facility (RRID:SCR_026393) for her advice and help with the use of the Spinning Disk 3i/Zeiss | Upright. We are grateful for use of and support from the following Paris Brain Institute's core facilities and their staff: iGenseq for RNAseq, Histomics for histology and ICM.Quant for imaging. We thank Dr. Wim Vandenberghe for kindly providing mito-Keima flies. We thank Dr. Christian Lobsiger for help with Laser Microdissection, all members of the Hassan lab for helpful discussions and Dr. Nicolas Renier for comments on the manuscript. We gratefully acknowledge Dr. Mounia Lagha for her support and guidance during the revision experiments, which were conducted in her laboratory following I. Mohylyak transfer to her group.

## Author contributions

I.M. and B.A.H. conceived the study, designed the experiments, and wrote the manuscript. I.M. performed *Drosophila* experiments and data analysis. M.A. conducted axon development and dendritic mitochondria evaluation experiments and data analysis. M.B. conducted behavioral experiments and data analysis. Z.K.A., M.D.W., and S.A. performed RNAseq analysis. C.P.C. and J.D.J.S. performed all hippocampal neuron experiments and analysis. N.A. and O.C. performed all COS7 cell experiments and analysis. N.D., M.C., C.M., and S.F.T. provided technical assistance. I.M., M.A., M.C., C.P.C., J.D.J.S., and B.A.H., performed revision experiments and revised manuscript.

## Competing interests

The authors declare no competing interests.
