## [Transparent Peer review file · Nature Communications]

Temporal transcriptional regulation of mitochondrial morphology primes activity-dependent circuit connectivity

Corresponding Author: Dr Bassem Hassan

Version 0:

Reviewer comments:

Reviewer #1

(Remarks to the Author)

The manuscript by Mohylyak and colleagues describes a newly discovered developmental transcription factor in *Drosophila*, TZAP/dZap. The telomere zinc finger-associated protein (TZAP) a time-dependent developmental regulator of expression of genes involved in mitochondrial biogenesis in developing neurons. I am not entirely convinced that the authors state that mitochondrial quality control genes (including Pink1) are heavily regulated. But I want to leave that for later discussion. The authors find that in *Drosophila*, loss of dTzap function during a critical developmental window leads to loss of activity-dependent synaptic connectivity, which can be rescued by Pink1 expression. At the cellular level, loss of dTzap/TZAP leads to changes in mitochondrial morphology and reduced neurotransmitter release in both fly and mammalian neurons. Remarkably, transient developmental downregulation of dTzap or Pink1 is sufficient to induce long-term changes in mitochondrial morphology, eliminate neuronal connectivity, and affect fly behavior.

This is a well-designed study with rigorous methodology. It seeks to answer an important question: how do neurons regulate mitochondria in axons given known synaptic connectivity during development? And TZAP appears to be the key factor. This study should have a wide audience. However, the current data provided cannot form a complete chain of evidence, so I cannot support the publication of the current manuscript. I hope the authors can address the concerns in the revised manuscript.

Main concerns:

- 1) A technical concern is that, based on the distribution of DCNs, it is difficult to isolate DCN clusters specifically using laser microdissection. What methods (or controls) did the authors use to demonstrate that there is no contamination from other cells? I understand that the majority of the sample can be DCN cells, but sometimes, a small number of non-target cells may also cause changes in the transcriptome.
- 2) The second technical concern is the dTzap RNAi line #100127 KK line from VDRC. 25% of this RNAi collection suffers from an unwanted insertion site causing ectopic expression of the Tiptop transcription factor. Crossing with one of the GAL4 driver lines induces false-positive phenotypes. These stocks can still be used, however, the researchers are responsible for cleaning up the stock (see Visser et al., Nature Communications 2016 and Green et al., Nature Methods).
- 3) The third technical concern is that the authors used a temperature-sensitive tub-Gal80ts repressor to adjust the expression of dTzap. However, it is not known whether temperature affects normal neural development. More precise control groups or the use of the gene-switch system are needed.
- 4) I understand that the authors want to find new downstream regulatory genes, but the contribution of MTFP1 cannot be ignored. Especially since it can regulate the balance of mitochondrial fission and fusion. So the question here is, does it have similar effects on mitochondrial morphology, DCN cell development, and synaptic connections? If not, how to explain it? If yes, does it work alone? What percentage of the TZAP phenotypic changes does it contribute? Does it work together with PINK1? Is it upstream or downstream of PINK1? These are all questions that need to be returned.
- 5) I have a question about the ability of TZAP to regulate mitochondrial morphology. The authors say that the downregulation of dTzap does not affect mitochondrial morphology in adult flies (line 312). However, in subsequent

mammalian cells COS7 experiments, mitochondrial morphology can be regulated by TZAP. How should we understand this phenomenon?

6) The author claimed that TZAP regulates the mitochondrial quality control pathway. However, only Pink1 was listed as an example. How about the other genes in the MQC pathway, such as p62, Marf, other Atg genes, especially Parkin? How does Parkin genetically interact with dTzap? Does it also change the morphology of mitochondria in dTzap mutants? Do they also change the fly behavior shown in Figure 6?

7) What about other factors in the mitochondrial biogenesis pathway, such as PGC1a and other nucleus genome-encoded mitochondrial proteins?

Reviewer #2

(Remarks to the Author)

This manuscript by Iryna Mohylyak and collaborators identified a novel role for the transcription factor Tzap in axon development in the drosophila visual system. Tzap was identified from a genetic screen and based on sequence homology with the human Telomeric Zinc-finger Associated Protein TZAP. Here the authors specifically link Tzap function in circuit connectivity with the regulation of mitochondrial genes, including the kinase Pink1. Rescue of Tzap phenotype by Pink1 expression demonstrates causality.

Overall, the manuscript presents convincing data and identified a previously unknown function of TZAP in axonal development. This is significant because TZAP is largely studied as a regulator of telomere length, so finding a function in neurons is unexpected. My opinion is that this manuscript is well suited for Nature communications and I am strongly favorable for publication. I have a few specific comments listed below that don't decrease my overall enthusiasm for this paper.

Specific comments

- 1) The identification of dTzap is based on homology with the human protein. This should be better detailed in the paper. Actually, the whole identification of sTzap is repeated at 3 instances in the manuscript: last sentence of page 3, first sentence of page 6, and first sentence of page 9. I would suggest to regroup the parts on CG7101 identification, make clear how it aligns with the human counterparts (a schema would be useful to complete supplementary figure 3 and showing full protein structure and predicted functional domains). The rationale to name CG7101 as TZAP rather than PRDM5 or ZNF497 (the latter having a higher percentage of similarity/identity) is not clear. Could CG7101 phenotypes be rescued by re-expression of human TZAP to show some degree of functional similarity? Likewise, what about PRDM5 and ZNF497?
- 2) I didn't get how the authors identified/selected mitochondrial proteins from TZAP CHIP-seq data. Are the six proteins listed in Figure S4A the only mitochondrial genes found in the CHIP-seq? Are mitochondrial proteins enriched overall?
- 3) The expression data presented in Figure S4 are not very convincing, despite evidence at the morphological and functional level. Could there be any other regulatory factor involved, and/or indirect effects of TZAP inhibition of the expression of those mitochondrial genes (for example, could we imagine that TZAP regulates the expression of a transcription factor with a broad effect on mitochondrial gene expression such as PGC1a)?
- 4) Does Tzap inhibition affect axonal and dendritic mitochondria morphology or is there some specificity for the axon?
- 5) The time-dependent regulation of Tzap using the inducible system is very elegant and strongly suggests a critical window during development when Tzap regulates mitochondria size. As it is known that mitochondria properties such as trafficking change over the course of axon development and neuronal maturation, and that size increase could make it more difficult to affect mitochondria at the extremity of the axon, could an explanation be that when Tzap is inhibited in the adult, it is too late to impact the morphology of mitochondria in the axon? This point could be answered in the discussion but does not need additional experiments.
- 6) In Cos7 cells, does TZAP affect cell cycle progression and if so, what is the impact on mitochondrial morphology?
- 7) The part on hippocampal neurons seems a bit underdeveloped. Is there an effect of TZAP inhibition on synaptic mitochondrial function (ATP production or calcium buffering)? This would nicely complement the glutamate release data.

Minor comments

- 1) The formatting of the references is sometimes odd, see for example references 12, 13 or 16. This must be a mistake from the software used to format references including the local file name to the DOI
- 2) Data presented are sometimes very small and may be hard to read
- 3) Cumulative frequency graphs could be better suited to present data than violin plot, as it makes it easier to visualize differences. This is a suggestion, the choice is left to the authors

Reviewer #4

(Remarks to the Author)

The authors laser-dissected samples for transcriptomic analyses and performed genetic screens, live imaging and behavioral assays to identify novel transcription factors that are essential for neuronal development. Dorsal cluster neurons (DCNs) were genetically labeled and laser-dissected for RNA-seq analyses to identify DCN-enriched gene expression. A genetic screen was subsequently performed and identified multiple transcription-related regulators including dTzap that regulate neuronal circuit formation during development. dTzap was further found to regulate mitochondria-related genes, and loss of dTzap led to morphological defects of mitochondria, synaptic connectivity defects and aberrant behaviors in flies. The mitochondria defects were also observed in human cell culture and rat primary neurons depleted for TZAP, which

displayed additional transmitter release defects. Furthermore, authors identified a key mitochondria gene Pink1, loss of which also led to mitochondria and connectivity defects, and importantly, Pink1 expression strongly rescued mitochondria and connectivity defects in loss of dTzap flies. The authors proposed that dTzap acts as a transcriptional regulator of mitochondria genes during neuronal development and facilitates neuronal activities and connectivity.

This story would be of interest to a broad range of scientists who study neuroscience, development, mitochondria, gene expression and neuro-circuit formation. The mechanisms identified in this manuscript reveal a novel dimension of TZAP function during development, which was previously studied for telomere regulation in the context of cancer. The identified key target Pink1 also implies essential mitochondria functions during development that may provide novel link between development and degenerative diseases.

One major concern is that the defects during neuronal development need better temporal resolution – when target proteins become depleted after the temperature shift or CRISPR KO, and what specific developmental process is affected. In addition, the authors assumed that dTzap shares the same binding profiles in whole adult flies and developing neurons, proposing direct regulation of target gene expression in DCNs. This assumption is not necessarily true. A third major concern is the interpretation of mitochondria defects – it requires further evidence to conclude that the defects are larger mitochondria sizes rather than fission/fusion defects. Additional experiments and revision to the text are needed to improve the manuscript.

Major:

1. I highly appreciate the consideration of transcriptomic specificity, but the samples collected are not cell type-specific since there was no cell sorting by flow cytometry, magnetic-based selection or manual selection after dissociation. LMD facilitates cell type-enrichment, and there are other cell types in collected samples.

2. The control samples in RNA-seq analyses were “randomly cut” brain regions, and these control replicates likely set very different references for downstream differential analyses. Notably, authors did clustering analyses to retain/exclude replicates. Therefore, selecting controls generated from these random regions determines what DE genes will be called. It is unclear how authors determine which replicates to retain/discard since replicate N15 is in the same control cluster Supplementary File 1 tab 1, and multiple other DCN replicates are clustered next to/closer to the red replicates.

3. Authors often showed axonal phenotypes in adults. If conclusions are development-related, it would be important to specify which specific developmental process is aberrant – initial growth, remodeling (pruning/regrowth), or axon maintenance during development by examining DCN axons at different stages during development. For example, some DCNs have axon innervation at correct targets – do other DCNs develop normal axons earlier but lose them later during development or they never grow axons at the first place? Do they survive without axons or undergo cell death? Tests at earlier developmental stages are essential to interpret the defects as growth, remodeling or maintenance defects during development.

Related temporal resolution is also essential to support the “initiation” conclusion in lines 268-269, which are currently based on adult brains.

4. Authors employed delicate temporal RNAi as well as CRISPR, which depend on cellular degradation of endogenous proteins (typically slow). The noted time window has the DNA-KO or transcript knockdown events, which do not necessarily affect protein functions until maybe much later. Authors have the tool dTzap-GFP to specify when dTzap proteins are actually down-regulated, and the conclusions may be shifted to a later developmental stage.

5. In line 252, it is unclear what tissues were used for ChIP-seq analyses (likely whole adult flies based on line 871). Considering the extremely low expression of dTzap in adult DCNs, it is not convincing to assume whole adult flies share the same binding profiles of dTzap with developing DCNs. There are multiple techniques that may determine dTzap binding profiles in DCNs, e.g., ChIP-seq/qPCR, CUT&Run or CUT&Tag. CUT&RUN and CUT&Tag have been established to work at single-cell levels, and can be applied to LMD-isolated tissues. At the least, the binding profiles can be determined with 10-15 dissected pupal brains, which are still way better than whole fly profiles at the adult stage. If there are no dTzap antibodies, anti-GFP can be used to probe dTzap-GFP.

6. Since authors concluded that dTzap regulates neuronal development, it is important to show that dTzap regulates mito genes during development (e.g., P30-P48) rather than adults – Figure S4C-E show critical data that are worth tests in developing brains as a regular figure.

7. Line 302, and Figures 3 and 4, the elongated mitochondria phenotypes may reflect incomplete/paused fission or ectopic fusion defects rather than elongation defects. Those possibilities may be ruled out by staining for various mito markers, e.g., Dnm1.

8. Starting from line 425, figure call-out is incorrect. Is there any morphological phenotype in rat neurons? How should we interpret presynaptic release defects – are they consequences of mito phenotypes/aberrant metabolism in neurons?

9. If the conclusion is that presynaptic sites are formed normally (Line 448), it is unclear why syt1-Cherry is much weaker in Figure 5B-B'. In addition, the presynaptic sites all look smaller in KD and the background signal is lower too in 5B. Since this is DCN-specific KD of dTzap, pan-neuronal presynaptic sites should not be affected. If the genetic KD and synapse labeling are different thus can explain the difference, please clarify it in the text.

Minor:

1. It is unclear what stage was targeted to collect RNA-seq samples in Figure 1.
2. Line 99 - authors mentioned the same kit with different names, SMARTseq2 vs. SMART-Seq v4 in Methods.
3. It seems most if not all track screenshots are from UCSC rather than IGV as authors stated in Methods. If so, please modify Methods. In Figure 1C, all tracks should have the same scale to conclude that ato is highly expressed in DCN samples but not CB samples. Same for the house keeping gene tracks, which do not currently have scales, which will help people understand their "similar" expression levels.
4. Supplementary File 1 tab 2 shows filtered genes for DE analyses, not DCN-expressing genes as the label said. It would be helpful if authors show the full gene list before filtering as well that typically shows >12-14k detected genes in fly RNA-seq, and that information will further validate the library quality – the direct output from DESeq2 will serve that purpose well.
5. It is hard to compare Tomato to DIC to tell the difference before and after LMD. It would be helpful to show DIC before and after LMD, and the same for Tomato. In Figure 1B, the dissected samples are ambiguously weak for Tomato – if there is photo-bleaching and authors intend to show the Tomato-positive samples after LMD isolation, IF staining may be needed.
6. It is not clear why authors did not comment on L-DCN phenotypes since they are the majority DCNs. If M-DCNs facilitate easier quantification, it is valuable information about what happens to L-DCNs and it seems there are strong phenotypes (Figures 1, 2 and S2).
7. Please reveal the details/identity of the control KK30B40D.
8. Overexpression of dTzap does not affect mito genes – this is interesting but there are often cases where transcripts can be overexpressed but not proteins – is it possible to show the OE of proteins? If not, authors may include this possibility in Discussion.
9. Additional assays are needed to validate aberrant mitochondria functions despite the morphological defects.
10. Line 375, "a" factor
11. Line 354, if COS7 cells are already stressed and showing phenotypes, it is unclear whether they are good models for additional tests – the final phenotypes could be a synthetic phenotype instead of a mito phenotype. Please clarify.
12. Figure 6B – seems Drp1 depletion leads to mito phenotypes in current images – maybe another representative image?
13. line 442, only half of DCNs innervate – do the other half of DCNs die or stay alive?
14. Line 459, current data cannot determine the causality between loss of activity vs. connectivity unless loss of activity occurs first. In addition, the loss of postsynaptic sites can be solely caused by loss of presynaptic release – if the synapses are not used, neurons remove them. It is actually surprising that some presynaptic sites are still established and maintained, considering the presynaptic release is blocked.
15. Line 783, "25 clusters/sample" means 25 clusters were collected from one section or one animal?
16. Line 787, please detail how the protocol was adapted.
17. Line 799, trimming is needed.
18. Line 802, please include what kind of scaling analyses were performed to exclude samples.
19. Line 825, please indicate what kind of primer sequences were trimmed for sequencing-through reads.
20. Line 830, typo fond. Plus, phantom peaks have been annotated and identified in black/grey lists for various species. Authors may intersect their peak calls with black/grey lists to identify and remove those phantom peaks. Removing KO peaks is not a standard method to analyze ChIP-seq. KO samples may show peaks if there are residue target proteins.
21. Line 835, add (Ref).
22. Line 874, please add reference or include details about how those programs perform alignment beyond visualization.
23. Line 368, it would be helpful if better figure labeling is included to reflect GCaMP.
24. In figure 6, some genes are required for mito development but not all genes – how efficient are those RNAi strains? Are they all strong?
25. The causality or the order of phenotypes may not be authors' scope in this paper but may be interesting to the field – which occurs first – mito phenotypes, activity defects and connectivity defects? The order may imply the causality. This is not necessary for revision but may be of authors' interest.

Version 1:

Reviewer comments:

Reviewer #1

(Remarks to the Author)

The authors have addressed my concerns. I have no additional comments. I will support publication.

Reviewer #2

(Remarks to the Author)

The authors have answered satisfyingly to my comments. Congratulations on this very elegant work.

Reviewer #4

(Remarks to the Author)

My concerns are mostly addressed. I suggest some text rephrasing.

Major:

1. Two reviewers raised the same concern about cell-type specificity of the transcriptomic data. If authors choose to emphasize such specificity, strong validation is required to demonstrate the presence of DCN cells and the absence of non-DCN cells. All information authors provided indicates the presence of DCN cells, but no information about markers of non-DCN cells, e.g., glial cells, is presented. The presence of DCN marker genes does not promise the absence of contamination. It is inappropriate to conclude about cell-type specificity without further validation.

Indeed, identifying cell type-specific mechanisms is challenging but definitely feasible. FACS has been successfully used to isolate similarly rare neuroblasts (~200 per brain, PMID: 23660757; PMID: 22884370) for multiomics analyses with solid validation of specificity, as well as the extremely rare LNvs neurons (8 per brain, PMID: 30380410) and gamma neurons (PMID: 30300589). If no additional assays are performed, I recommend authors to tune down their conclusions about “cell type-specific” transcriptome.

** Based on supplementary file 1 tab 1, seven KD samples are used for analyses rather than six, line 101.

4. The text states, “Using an endogenously GFP-tagged version to examine...” so it implies that the endogenous Mirana is tagged by GFP. Authors responses suggested that the Mirana-GFP is a transgenic insertion, in conflict with the text. Please clarify and cite the transgenic fly strain.

Even if the Mirana-GFP is a transgenic insertion, it is important and a great tool to titrate Mirana protein degradation by monitoring GFP levels since RNAi targets both endogenous and transgenic Mirana RNAs. The point is, when does the Mirana protein function during development? RNAi was induced between L3 and APF48 to knockdown RNAs, but protein levels may stay unchanged, therefore it is arguable to conclude about temporal Mirana functions solely based on when RNAs are degraded.

This is a great manuscript proposing precise temporal resolution during development. I suggest at least discuss such possible RNA/protein discordance in Discussion if no additional experiments are performed.

** is it a typo in line 153 “at an earlier developmental stage (P48)” since authors just mentioned earlier that loss of Mirana between L3 and P48 leads to defects, but then P48 has not difference?

5. Please make it clear embryo CHIP-seq was used.

Minor:

17. I meant reads need to be trimmed for sequencing quality and adapter sequences before alignment for RNA-seq analyses, line 759.

19. This command does not trim adapter sequences, unless -a/-A or -g/-G is used.

We thank the reviewers for their positive comments on the conceptual advance and breadth of interest of the work, and for their helpful remarks and suggestions. We have addressed their comments with new experiments that strengthened and further extended our discovery. In particular, we extend our findings to show the impact of TZAP loss on mitochondrial function. This has resulted in a slight change to the title of the manuscript which is now “**Temporal transcriptional regulation of mitochondrial biology primes activity-dependent circuit connectivity**”. Furthermore, we have also renamed the *Drosophila* functional homologue of human TZAP as “*Mirana*”.

Below we address the specific comments by each reviewer in a point-by-point response letter. For the convenience of the reviewers, we have included updated figures with new data in this response letter in addition to indicating the **new figure numbers and pages** in the revised manuscript. We have also included a version of the revised manuscript with tracked changes and **highlighted** new sections for ease of review.

Reviewer #1 (Remarks to the Author):

This is a well-designed study with rigorous methodology. It seeks to answer an important question: how do neurons regulate mitochondria in axons given known synaptic connectivity during development? And TZAP appears to be the key factor. This study should have a wide audience. However, the current data provided cannot form a complete chain of evidence, so I cannot support the publication of the current manuscript. I hope the authors can address the concerns in the revised manuscript.

We thank the reviewer for their positive comments on the study and its importance and rigor, and address their specific concerns below. Please note that based on reviewer 2 comments we have now renamed the protein encoded by the fly gene CG7101 as Mirana (Mitochondrial integrity regulator of neuronal architecture). We therefore refer to it as Mirana in the rest of this letter and the revised manuscript.

Main concerns:

1) A technical concern is that, based on the distribution of DCNs, it is difficult to isolate DCN clusters specifically using laser microdissection. What methods (or controls) did the authors use to demonstrate that there is no contamination from other cells? I understand that the majority of the sample can be DCN cells, but sometimes, a small number of non-target cells may also cause changes in the transcriptome.

We used several criteria to ensure that the transcripts we examined were really expressed in DCNs:

- 1- We isolated control areas surrounding, but excluding, the DCNs to make sure that any contamination of non-DCN cells within the DCN cluster are also represented in control samples
- 2- We compared the expression of two key DCN marker genes, *ato* and *acj6* in DCN clusters and control clusters. This showed that both were expressed in DCN clusters but not control clusters
- 3- We performed enrichment analysis to make sure that we only consider genes whose expression in DCNs was not due to potential contamination from highly expressed genes in surrounding cells

To address the reviewer's concern, we have now compared our gene set to those proposed by *Özel et al., 2020* scRNAseq data as DCN cluster markers. We find that our dataset contains 134 of the 135 genes identified as DCN markers. We now show these comparisons in **Supplemental Figure S3** (included below for the convenience of the reviewer) and state them on **page 6** of the revised manuscript. This is further strongly supported by the finding that 10 of the 74 genes we find to be DCN-expressed cause medulla axon innervation phenotypes, demonstrating that our approach identified genes that are both expressed during and required for DCN development.

Response letter figure 1: DCN marker genes in our transcriptomic data set

2) The second technical concern is the dTzap RNAi line #100127 KK line from VDRC. 25% of this RNAi collection suffers from an unwanted insertion site causing ectopic expression of the Tiptop transcription factor. Crossing with one of the GAL4 driver lines induces false-positive phenotypes. These stocks can still be used, however, the researchers are responsible for cleaning up the stock (see Visser et al., Nature Communications 2016 and Green et al., Nature Methods).

We are aware of this issue which is why we used two independent RNAi lines (VDRC_KK100127 and VDRC_GD27849) and a CRISPR guide RNA, thus mitigating off-target effects. Importantly, as a control in all our experiments, we used KK^{30B40D}, the host strain for the KK library (VDRC_ID 60100). This line contains landing sites at both 40D and 30B, thus mitigating the site insertion effect. Given the similarity of phenotypes across our RNAi and CRISPR lines and the absence of these phenotypes in the landing site control lines, we are very confident that the phenotypes we report are specific.

3) The third technical concern is that the authors used a temperature-sensitive tub-Gal80ts repressor to adjust the expression of dTzap. However, it is not known whether temperature affects normal neural development. More precise control groups or the use of the gene-switch system are needed.

All control experiments were carried under the exact same temperature conditions using the insertion site control lines, thus mitigating any potential temperature-related effects. We apologize if this was not clear in the original manuscript. We have now made this explicitly clear in the methods section on **page 35** of the revised manuscript.

4) I understand that the authors want to find new downstream regulatory genes, but the contribution of MTFP1 cannot be ignored. Especially since it can regulate the balance of mitochondrial fission and fusion. So, the question here is, does it have similar effects on mitochondrial morphology, DCN cell development, and synaptic connections? If not, how to explain it? If yes, does it work alone? What percentage of the TZAP phenotypic changes does

it contribute? Does it work together with PINK1? Is it upstream or downstream of PINK1? These are all questions that need to be returned.

We thank the reviewer for raising this important issue. Addressing it has allowed us to expand and strengthen our findings. We agree with the reviewer that MTFP1 is a very important target, based on work from the Kappei lab (Jahn et al., 2017). We had tested the contribution of MTFP1 to the mitochondrial morphology phenotype and were surprised to find that that knock-down of *dMtfp1* did not cause the same enlarged presynaptic mitochondria as the knock-down of *Mirana*, suggesting it does not contribute to this aspect of the phenotype (**Figure 6E of the original manuscript**).

However, based on the reviewer's request, we asked whether knock-down of *dMtfp1* phenocopies and any other aspect of the *Mirana* phenotype. We found that it causes a reduction in the number of DCN axons in the medulla. We have now added these data in Supplemental **Figure S9** (included below for the convenience of the reviewer) and described **on page 28** of the revised manuscript. This lead us to postulate that mitochondrial fission genes downstream of *Mirana* were specifically required to regulate axonal growth, whereas mitochondrial quality control was required for synaptic connectivity. To test this idea, we examined medulla innervation upon knockdown of *Drp1*, a key mitochondria fission gene. RNAi against *Drp1* in DCNs showed very similar decrease of medulla innervation to *Mirana*-RNAi (**Figure S9**; included below for the convenience of the reviewer). In contrast, knock-down of *Drp1* also did not cause enlargement of synaptic mitochondria. Based on these new data, we added the following sentence on **page 26**: "Although the regulation of synaptic mitochondria and the resulting activity-dependent connectivity by *Mirana* are mediated by the mitochondrial quality control genes like *Pink1* and *Parkin*, target genes regulating mitochondrial fission, notably *Mtfp1* and *Drp1*, appear to mediate the role of *Mirana* in ensuring a sufficient number of DCN axons in the medulla."

Response letter figure 2. CG7772/MTFP1 knockdown reduces medulla axon number but not enlarged presynaptic mitochondria. *Drp1* knockdown reduces medulla axon number but not enlarged presynaptic mitochondria.

5) I have a question about the ability of TZAP to regulate mitochondrial morphology. The authors say that the downregulation of *dTzap* does not affect mitochondrial morphology in adult flies (line 312). However, in subsequent mammalian cells COS7 experiments, mitochondrial morphology can be regulated by TZAP. How should we understand this phenomenon?

As shown by the GFP-fusion reporter, the major expression of Mirana is during a narrow developmental window. There appears to be very low Mirana protein expression in adult DCNs. Thus, it is not surprising that the main effects of Mirana loss of function coincide with its period of protein expression during development, in these postmitotic neurons. This maybe different in COS7 cells which are actively dividing cells. Importantly, the situation in DCN reflects an *in vivo* situation whereas COS7 cells reflect an *in vitro* situation. Nevertheless, the fact that both Mirana and TZAP cause mitochondrial morphology defects across systems strongly supports a conserved function.

6) The author claimed that TZAP regulates the mitochondrial quality control pathway. However, only *Pink1* was listed as an example. How about the other genes in the MQC pathway, such as *p62*, *Marf*, other *Atg* genes, especially *Parkin*? How does *Parkin* genetically interact with *dTzap*? Does it also change the morphology of mitochondria in *dTzap* mutants? Do they also change the fly behavior shown in Figure 6?

7) What about other factors in the mitochondrial biogenesis pathway, such as *PCG1a* and other nucleus genome-encoded mitochondrial proteins?

We thank the reviewer for this excellent suggestion. We performed RNAi depletion of *parkin*, *srl(PCG1)* and *Marf1* in DCN neurons. We find that while that *srl/PCG1* and *Marf1* downregulation had no effect on mitochondrial size, *park*-RNAi caused the same phenotypes as Mirana and *Pink1*, further confirming the specific role of *Pink1*-*Parkin* dependent mitochondrial quality control downstream of Mirana in mitochondrial regulation of synaptic connectivity. We have included these data in **Figure 6B, E** (included below for the convenience of the reviewer) and described them on **page 26** of the revised manuscript. These new findings strongly support the regulation of the mitochondrial quality control pathway by Mirana. We believe that the rescue of all Mirana morphological phenotypes by *Pink1* constitutes sufficient evidence for the genetic connection between Mirana and the *Pink1* pathway. While we appreciate that testing each additional target gene and including further behavioral experiments is in principle interesting, it is highly unrealistic, especially given the complexity of the genotypes that need to be tested and considering that it will not add fundamental insight to the discovery of the role of Mirana in neuronal circuit development.

Response letter figure 3. Knockdown of mitochondrial quality control genes phenocopies the mitochondrial size increase observed upon *Mirana* knockdown.

Reviewer #2 (Remarks to the Author):

My opinion is that this manuscript is well suited for Nature communications and I am strongly favorable for publication. I have a few specific comments listed below that don't decrease my overall enthusiasm for this paper.

We thank the reviewer for their positive comments and their support of publication and address their specific concerns below. Please note that based on the reviewer's first comment we have now renamed the protein encoded by the fly gene CG7101 as Mirana (Mitochondrial integrity regulator of neuronal architecture). We therefore refer to it as Mirana in the rest of this letter and the revised manuscript.

Specific comments

1) The identification of dTzap is based on homology with the human protein. This should be better detailed in the paper. Actually, the whole identification of dTzap is repeated at 3 instances in the manuscript: last sentence of page 3, first sentence of page 6, and first sentence of page 9. I would suggest to regroup the parts on CG7101 identification, make clear how it aligns with the human counterparts (a schema would be useful to complete supplementary figure 3 and showing full protein structure and predicted functional domains). The rationale to name CG7101 as TZAP rather than PRDM5 or ZNF497 (the latter having a higher percentage of similarity/identity) is not clear. Could CG7101 phenotypes be rescued by re-expression of human TZAP to show some degree of functional similarity? Likewise, what about PRDM5 and ZNF497?

We initially used this name to indicate the functional similarity we discovered *a posteriori*. We agree with the reviewer that there was no *a priori* reason to name CG7101 as dTzap. Based on this, **we have now given the fly gene a new, independent, name – Mirana** – although we retain the functional comparison in the manuscript, because we believe the similarity in phenotypes is both interesting and instructive given the similarity of effects on mitochondria (also see new data below) between CG7101 KD in flies and TZAP KD in rat neurons. As stated in the original manuscript, we started out by testing functional similarity of TZAP, simply because more was known about it, while the other two genes are not studied functionally to our knowledge. As requested, we have regrouped the sentences referring to the similarity between CG7101 and TZAP on **page 11** of the revised manuscript. The sentence at the end of page 3 only refers to the fly gene and does not yet make the comparison, we therefore kept it there. The rescue experiments with mammalian TZAP would be in principle interesting but likely to be complicated by the BTB domain that does not exist in CG7101, and we feel doing domain identification is beyond this work.

2) I didn't get how the authors identified/selected mitochondrial proteins from TZAP CHIP-seq data. Are the six proteins listed in Figure S4A the only mitochondrial genes found in the CHIP-seq? Are mitochondrial proteins enriched overall?

We initially focused on selected mitochondrial genes identified as TZAP targets in human cell lines, as referenced in the original manuscript, based on the work of Kappei lab (Jahn et al., 2017). However, based on the reviewer's comment, we performed both GO analysis of available CG7101/Mirana ChIPseq data and manual curation for other mitochondrial genes

with significant peaks. While we found no GO term enrichment for mitochondrial quality control, manual curation revealed peaks around many other important mitochondrial genes, including Parkin. We have now included a **Supplementary Table 2** in the revised manuscript listing these genes (included below for the convenience of the reviewer), as well as several key mitochondrial genes around which we found no peaks, including PCG1.

	Peaks found				No peaks			
	Gene name Drosophila	Human homolog	Function	Number of peaks by gene	Gene name Drosophila	Human homolog	Function	Number of peaks by gene
Mitochondrial translation	CG7303	TZAP	Transcription factor	1	mtpe1	MRPL1		1
	mtit	TRAK2/TRAK2	Mitochondria transport	6	mtpe10	MRPL10		1
	Miro	RHOT2/RHOT2	Mitochondria transport	1	mtpe11	MRPL11		1
	Dp1	DRP1	Mitochondrial fission	1	mtpe13	MRPL13		1
	ewp	MRP1	Mitochondria biogenesis	2	mtpe16	MRPL16		1
	Mef	MFN2/Z	Mitochondrial fusion	1	mtpe17	MRPL17		1
	Matqin	JMJD1/MEK2	Cristae architecture	2	mtpe18	MRPL18		2
	osc/Nox2	NFE2L1/NFE2L2/NFE2L3	oxidative stress genes regulation	6	mtpe19	MRPL19		1
	Tom20	MTOM20	Mitochondrial import	1	mtpe2	MRPL2		1
	park	PARK1	Quality control	1	mtpe20	MRPL20		1
	Plk2	PINK1	Quality control	1	mtpe21	MRPL21		1
Mitochondrial dynamics					mtpe23	MRPL23		2
					mtpe24	MRPL24		1
					mtpe27	MRPL27		3
Autophagy and quality control					mtpe28	MRPL28		1
					mtpe32	MRPL32		1
					mtpe33	MRPL33		1
					mtpe34	MRPL34		1
					mtpe35	MRPL35		1
					mtpe37	MRPL37		1
					mtpe39	MRPL39		1
mtDNA replication and maintenance					mtpe4	MRPL4		1
					mtpe41	MRPL41		1
					mtpe43	MRPL43		1
					mtpe44	MRPL44		1
					mtpe47	MRPL47		1
					mtpe48	MRPL48		1
					mtpe50	MRPL50		1
					mtpe51	MRPL51		1
					mtpe52	MRPL52		2
					mtpe53	MRPL53		1
					mtpe54	MRPL54		1
				mtpe55	MRPL55		2	
				mtpe9	MRPL9		1	
Mitochondrial transcription					mtpe42	MRPL42		1
					mtpe43	MRPL43		1
					mtpe44	MRPL44		1
Mitochondrial dynamics					mtpe46	MRPL46		1
					mtpe47	MRPL47		1
					mtpe48	MRPL48		1
					mtpe50	MRPL50		1
					mtpe51	MRPL51		1
					mtpe52	MRPL52		2
					mtpe53	MRPL53		1
					mtpe54	MRPL54		1
					mtpe55	MRPL55		2
					mtpe9	MRPL9		1
	Mitochondrial translation					mtpe41	MRPL41	
					mtpe43	MRPL43		1
					mtpe44	MRPL44		1
					mtpe47	MRPL47		1
					mtpe48	MRPL48		1
					mtpe50	MRPL50		1
					mtpe51	MRPL51		1
					mtpe52	MRPL52		2
					mtpe53	MRPL53		1
					mtpe54	MRPL54		1
					mtpe55	MRPL55		2

Response letter figure 4. Number of normalized peaks for CG7101-GFP ChIP-Seq data sets from whole female fly (ModEnCode project).

3) The expression data presented in Figure S4 are not very convincing, despite evidence at the morphological and functional level. Could there be any other regulatory factor involved, and/or indirect effects of TZAP inhibition of the expression of those mitochondrial genes (for example, could we imagine that TZAP regulates the expression of a transcription factor with a broad effect on mitochondrial gene expression such as PGC1a)?

We note that Figure S4 shows a reduction of 80% of target gene expression upon RNAi knockdown which we consider to be quite significant. Nevertheless, the reviewer is correct in suggestion that we cannot exclude the involvement of another factor and PCG1 would have been an interesting and logical candidate. To test this, we first asked if Mirana binds to the srl/PCG1 locus, and found this not to be the case, as shown in **Supplementary Table 2** (included above for the convenience of the reviewer) of the revised manuscript. Next, we tested whether KD of srl/PCG1 phenocopies Mirana and also found this not to be the case. While these data do not rule out a potential unknown co-factor, they do suggest that Mirana is a major regulator of mitochondrial quality control genes during neuronal circuit development in flies.

4) Does *Tzap* inhibition affect axonal and dendritic mitochondria morphology or is there some specificity for the axon?

By expression dendritic marker Denmark, together with mito-GFP we were able to examine mitochondria size in DCN dendrites, and axons of L-DCNs, which we haven't studied before. Like in case of M-DCNs, we see significant elongation in both, dendrites and axons at optic chiasm region. This observation is ruling out specificity of Mirana requirements in particular neuronal compartment. We added this finding to the **Supplemental Figure S6** of revised manuscript, and described them on **page 15** of the revised manuscript.

Response letter figure 5. Mitochondria length quantification in L-DCNs dendrites and axons using Denmark marker shows elongation of mitochondria in both compartments in case of Mirana depletion.

5) The time-dependent regulation of *Tzap* using the inducible system is very elegant and strongly suggests a critical window during development when *Tzap* regulates mitochondria size. As it is known that mitochondria properties such as trafficking change over the course of axon development and neuronal maturation, and that size increase could make it more difficult to affect mitochondria at the extremity of the axon, could an explanation be that when *Tzap* is inhibited in the adult, it is too late to impact the morphology of mitochondria in the axon? This point could be answered in the discussion but does not need additional experiments.

We have added this point to the discussion on **page 31** of the revised manuscript.

6) In *Cos7* cells, does *TZAP* affect cell cycle progression and if so, what is the impact on mitochondrial morphology?

This is an interesting question. However, given that the impact on mitochondrial size in COS7 and neurons in the rat and the fly are similar, it is unlikely the cell cycle is a key determinant of the phenotype, and any potential impact on the cell cycle of COS7 cells would be different from the effects observed in neurons and is thus outside the scope of this work.

7) The part on hippocampal neurons seems a bit underdeveloped. Is there an effect of TZAP inhibition on synaptic mitochondrial function (ATP production or calcium buffering)? This would nicely complement the glutamate release data.

We have now examined the effects of TZAP KD on mitochondrial calcium dynamics using mito-GCaMP6f and ATP production using the cytosolic ATP sensor, iATPSnFR2 in rat neurons. We found a ~50% reduction in mitochondrial Ca^{2+} uptake and a more pronounced decrease in ATP levels after stimulation, suggesting that activity-driven mitochondrial ATP production could be less efficient in TZAP KD neurons and potentially explaining the reduction in glutamate release. These new data are shown in **Figure 4** (included below for the convenience of the reviewer) and described on **page 18** of the revised manuscript.

Response letter figure 6. Impact of TZAP knockdown on mitochondrial morphology and function in mammalian cells.

Minor comments

1) The formatting of the references is sometimes odd, see for example references 12, 13 or 16. This must be a mistake from the software used to format references including the local file name to the DOI

We have fixed the references.

2) Data presented are sometimes very small and may be hard to read

We tried to increase the size and font in embedded figures.

3) Cumulative frequency graphs could be better suited to present data than violin plot, as it makes it easier to visualize differences. This is a suggestion, the choice is left to the authors.
We prefer the violin plot presentation.

Reviewer #4 (Remarks to the Author):

This story would be of interest to a broad range of scientists who study neuroscience, development, mitochondria, gene expression and neuro-circuit formation. The mechanisms identified in this manuscript reveal a novel dimension of TZAP function during development, which was previously studied for telomere regulation in the context of cancer. The identified key target Pink1 also implies essential mitochondria functions during development that may provide novel link between development and degenerative diseases.

We thank the reviewer for their positive comments on the significance of our discovery and address their specific concerns below. Please note that based on reviewer 2 comments we have now renamed the protein encoded by the fly gene CG7101 as Mirana (Mitochondrial integrity regulator of neuronal architecture). We therefore refer to it as Mirana in the rest of this letter and the revised manuscript.

Major:

1. I highly appreciate the consideration of transcriptomic specificity, but the samples collected are not cell type-specific since there was no cell sorting by flow cytometry, magnetic-based selection or manual selection after dissociation. LMD facilitates cell type-enrichment, and there are other cell types in collected samples.

We agree with the reviewer and note that even FACS does not guarantee absolute purity and performing FACS on such a small population of postmitotic fly neurons is very difficult.

In order to ensure a high degree of specificity, we used several criteria to ensure that the transcripts we examined were really expressed in DCNs:

- 1- We isolated control areas surrounding, but excluding, the DCNs to make sure that any contamination of non-DCN cells within the DCN cluster are also represented in control samples
- 2- We compared the expression of two key DCN marker genes, *ato* and *acj6* in DCN clusters and control clusters. This showed that both were expressed in DCN clusters but not control clusters
- 3- We performed enrichment analysis to make sure that we only consider genes whose expression in DCNs was not due to potential contamination from highly expressed genes in surrounding cells

To address the reviewer's concern, we have now compared our gene set to those proposed by *Özel et al., 2020* scRNAseq data as DCN cluster markers. We find that our dataset contains 134 of the 135 genes identified as DCN markers. We now show these comparisons in Supplemental **Figure S3** and state them on **page 8** of the revised manuscript. This is further strongly supported by the finding that 10 of the 74 genes we find to be DCN-expressed cause medulla axon innervation phenotypes, demonstrating that our approach identified genes that are both expressed during and required for DCN development.

2. The control samples in RNA-seq analyses were "randomly cut" brain regions, and these control replicates likely set very different references for downstream differential analyses. Notably, authors did clustering analyses to retain/exclude replicates. Therefore, selecting controls generated from these random regions determines what DE genes will be called. It is unclear how authors determine which replicates to retain/discard since replicate N15 is in the

same control cluster Supplementary File 1 tab 1, and multiple other DCN replicates are clustered next to/closer to the red replicates.

We believe that fact that our transcriptome included essentially all the marker genes from later generated single cell data, as stated above, validates our selection of negative control regions. As for replicate N15, although it was well clustered, it was excluded as it was different from other samples in the way it was produced. This was a sample from single DCN cluster region that underwent direct lysis in contrast to the other samples which contained 20-25 DCN clusters each (20-25).

3. Authors often showed axonal phenotypes in adults. If conclusions are development-related, it would be important to specify which specific developmental process is aberrant – initial growth, remodeling (pruning/regrowth), or axon maintenance during development by examining DCN axons at different stages during development. For example, some DCNs have axon innervation at correct targets – do other DCNs develop normal axons earlier but lose them later during development or they never grow axons at the first place? Do they survive without axons or undergo cell death? Tests at earlier developmental stages are essential to interpret the defects as growth, remodeling or maintenance defects during development. Related temporal resolution is also essential to support the “initiation” conclusion in lines 268-269, which are currently based on adult brains

We thank the reviewer for raising this interesting point. To address this issue, we generated stochastic single cell clones and asked whether any of them showed an abnormal targeting outside either the medulla or lobula. We found this not to be the case, which means that all axons that failed to reach the medulla innervate the lobula. We have also compared axonal targeting at P48 (when DCN axons have begun to reach the medulla) versus adult and found that there is no difference at P48, meaning that the adult phenotype is due to the retraction of axons from the medulla back to the lobula after P48 but before eclosion. These data are now shown in **Supplementary Figure S4** (included below for the convenience of the reviewer) and described on **page 6 and 10** of the revised manuscript. In terms of the conclusion about the initial requirement of Mirana, we believe we have very strong temporal data in the manuscript supporting this conclusion (**Figure 3 of the original manuscript**). First, we performed a careful high temporal resolution of analysis of Mirana protein expression and showed that its peak of expression is between 30 and 50 hours of pupal development. Second, we performed developmental examination of the requirement of Mirana at relatively high temporal resolution and showed that it is required during the first 48 hours of pupal development, but not later, consistent with the peak of its expression. Third, we showed that the mitochondrial phenotype first emerges at 72hrs of pupal development about 20-30 hours after the peak of Mirana expression, and continues to deteriorate over time, even though Mirana itself is no longer expressed.

MOHYLYAK_Figure S4

Response letter figure 7. MCFO clones show no innervation of novel or erroneous areas by either L-DCNs or M-DCNs, and developmental analysis defines the axonal phenotype as a retraction phenotype.

4. Authors employed delicate temporal RNAi as well as CRISPR, which depend on cellular degradation of endogenous proteins (typically slow). The noted time window has the DNA-KO or transcript knockdown events, which do not necessarily affect protein functions until maybe much later. Authors have the tool *dTzap-GFP* to specify when *dTzap* proteins are actually down-regulated, and the conclusions may be shifted to a later developmental stage.

We believe this might stem from a misunderstanding. The DNA knockout with the guide RNA was performed well before Mirana is expressed in DCNs. This is because the guide RNA was driven using the *Ato-Gal4* which is expressed very early in post-mitotic DCNs. Thus, by the time Mirana would be expressed there is no expression possible, because of the knockout of the gene. The GFP-reporter is a genomic transgene driven by endogenous regulatory sequences, and thus would not be a readout for the knockout of the endogenous gene. The same is true

for the RNAi lines. As *Mirana* RNA is expressed before *Mirana* protein, knockdown of the RNA should prevent protein expression. Furthermore, the temporal developmental knock-down of *Mirana* only between L3 and P48, but not later cause the same phenotypes as prolonged knock-down. Thus, the effects cannot be due to a function at later time point, because we allow the expression of *Mirana* to be restored. Finally, we note that we did show endogenous knockdown of *Mirana* by qRT-PCR upon the RNAi (**Figure S4 of the original manuscript**).

5. In line 252, it is unclear what tissues were used for ChIP-seq analyses (likely whole adult flies based on line 871). Considering the extremely low expression of dTzap in adult DCNs, it is not convincing to assume whole adult flies share the same binding profiles of dTzap with developing DCNs. There are multiple techniques that may determine dTzap binding profiles in DCNs, e.g., ChIP-seq/qPCR, CUT&Run or CUT&Tag. CUT&RUN and CUT&Tag have been established to work at single-cell levels, and can be applied to LMD-isolated tissues. At the least, the binding profiles can be determined with 10-15 dissected pupal brains, which are still way better than whole fly profiles at the adult stage. If there are no dTzap antibodies, anti-GFP can be used to probe dTzap-GFP.

We used publicly available embryo ChIPseq data to identify potential targets. Given the functional validation of putative target genes and the rescue of the phenotypes by *Pink1*, we believe we have provided strong enough evidence for the validity of our approach. Performing such experiments on DCNs would be extremely challenging as there are no more than 80 DCNs per fly brain. While very interesting to try, identifying new potential *Mirana* target genes in DCNs is beyond the scope of this work.

6. Since authors concluded that dTzap regulates neuronal development, it is important to show that dTzap regulates mito genes during development (e.g., P30-P48) rather than adults – Figure S4C-E show critical data that are worth tests in developing brains as a regular figure.

We note that the target genes would be downregulated well after *Mirana* and thus not between P30-P48. While this is in principle interesting, performing qRT PCR on pupal brains is more difficult than on adults due to tissue sensitivity. Furthermore, we note that we demonstrated that *Pink1* is required between P48 and eclosion but not later, thus confirming the role of *Mirana* and its targets during development.

7. Line 302, and Figures 3 and 4, the elongated mitochondria phenotypes may reflect incomplete/paused fission or ectopic fusion defects rather than elongation defects. Those possibilities may be ruled out by staining for various mito markers, e.g., Dnm1.

We thank the reviewer for this interesting point. To address the role of fission/fusion, we had tested the contribution of MTFP1 to the mitochondrial morphology phenotype and were surprised to find that that knock-down of *dMtfp1* did not cause the same enlarged presynaptic mitochondria we as the knock-down of *Mirana*, suggesting it does not contribute to this phenotype (**Figure 6E of the original manuscript, Supplementary Figure S9 of revised manuscript**). Importantly, knock-down of *Drp1* also did not cause a mitochondrial phenotype in DCNs, suggesting that the phenotypes we observe are not primarily due to defects in mitochondrial fission (**Supplementary Figure S9 of revised manuscript**).

Next, we used mito-Keima to quantify mitophagy in DCN axon presynaptic terminals and found no difference between controls and *Mirana* knock-down DCNs. Given the evidence that *Mirana* and *TZAP* absence both lead to mitochondrial enlargement and reduced mitochondrial function, these data suggest that a failure in engaging additional mitophagy to remove

damaged mitochondria, consistent with downregulation of Pink1, Parkin and Atg1 upon Mirana knockdown. These new data are shown in **Supplementary Figure S10** (included below for the convenience of the reviewer) and described on **page 29** of the revised manuscript.

Response letter figure 8. Mitophagy is not activated in DCN axons upon Mirana knockdown.

8. Starting from line 425, figure call-out is incorrect. Is there any morphological phenotype in rat neurons? How should we interpret presynaptic release defects – are they consequences of mito phenotypes/aberrant metabolism in neurons?

We thank the reviewer for noting the figure call out error and have corrected it.

We have now examined the effects of TZAP KD on mitochondrial calcium dynamics using mito-GCaMP6f and ATP production using the cytosolic ATP sensor, iATPSnFR2 in rat neurons. We found a ~50% reduction in mitochondrial Ca^{2+} uptake and a more pronounced decrease in ATP levels after stimulation, suggesting that activity-driven mitochondrial ATP production could be less efficient in TZAP KD neurons and potentially explaining the reduction in glutamate release. These new data are shown in **Figure 4** and described on **page 18** of the revised manuscript (included above as figures 5 and 7 of this letter for the convenience of the reviewer).

9. If the conclusion is that presynaptic sites are formed normally (Line 448), it is unclear why syt1-Cherry is much weaker in Figure 5B-B'. In addition, the presynaptic sites all look smaller in KD and the background signal is lower too in 5B. Since this is DCN-specific KD of dTzap, pan-neuronal presynaptic sites should not be affected. If the genetic KD and synapse labeling are different thus can explain the difference, please clarify it in the text.

We are unsure what the reviewer is referring to. Syt-pHluorin is expressed specifically in DCNs in this experiment, thus there is no pan-neuronal synaptic signal in these images. The actual number of red puncta, representing the total number of pre-synaptic sites is the same between the genotypes as expected (see accompanying graph below). Finally, the only metric that is relevant is the ratio of the released vesicles (green) to the total number of vesicles (red) within the DCNs (where Syt-pHluorin is exclusively expressed), which is drastically reduced as shown in **Figure 5 of the original manuscript**.

Response letter figure 9. Comparison of the number of sites containing synaptic vesicles.

Minor:

1. It is unclear what stage was targeted to collect RNA-seq samples in Figure 1.

Adult fly brains were used for LMD samples. Now we mention this in Figure 1 description.

2. Line 99 - authors mentioned the same kit with different names, SMARTseq2 vs. SMART-Seq v4 in Methods.

We updated the kit name in line 100 of the manuscript.

3. It seems most if not all track screenshots are from UCSC rather than IGV as authors stated in Methods. If so, please modify Methods. In Figure 1C, all tracks should have the same scale to conclude that *ato* is highly expressed in DCN samples but not CB samples. Same for the house keeping gene tracks, which do not currently have scales, which will help people understand their “similar” expression levels.

In methods we state IGV usage for ChIPseq tracks visualization. Indeed, at Figure 1 screenshots for *ato* and RIL10 were made from UCSC. We add this comment to methods section. The scale is exactly the same for both conditions, *ato* and RPL10. We added scale to Rpl10 tracks into revised Figure 1C, and are attaching here uncropped screenshots from UCSC viewer.

4. Supplementary File 1 tab 2 shows filtered genes for DE analyses, not DCN-expressing genes as the label said. It would be helpful if authors show the full gene list before filtering as well that typically shows >12-14k detected genes in fly RNA-seq, and that information will further validate the library quality – the direct output from DEseq2 will serve that purpose well.

We edited the name of tab2 at File 1 to “Differentially expressed genes in DCNs”. We are providing all the raw fasta files of our RNAseq experiment, and they will be publically available.

5. It is hard to compare Tomato to DIC to tell the difference before and after LMD. It would be helpful to show DIC before and after LMD, and the same for Tomato. In Figure 1B, the dissected samples are ambiguously weak for Tomato – if there is photo-bleaching and authors intend to show the Tomato-positive samples after LMD isolation, IF staining may be needed.

RedStinger fluorescence in Figure 1B is used only for visualization purpose. We directly use cut tissue for lysis and RNA isolation so we do not see a compelling reason to fix and perform antibody staining of this marker. In our cryosections CD8::GFP green marker labelling DCN membranes, and RedStinger-Tomato marker labeling DCN cell bodies were used exclusively to visualize DCN clusters and help navigate the dissections. We added an image of dissected region viewed also in UV light (**Figure 1B_1'**), and increased contrast in dissected clusters for better visualization purpose (**Figure 1B_2, 2'**).

6. It is not clear why authors did not comment on L-DCN phenotypes since they are the majority DCNs. If M-DCNs facilitate easier quantification, it is valuable information about what happens to L-DCNs and it seems there are strong phenotypes (Figures 1, 2 and S2).

We thank reviewer for this point. To examine the impact of Mirana downregulation at single cell level we used advantage of Multi Color Flip-out system to stochastically label individual DCNs. Indeed, it is much easier to quantify M-DCNs, but with this labeling we can avoid crowding in lobula region and evaluate L-DCN axons morphology or any mistargeting effects and found that DCNs axons in the lobula show normal targeting and morphology. Next, we examined mitochondrial size in L-DCN axons and all DCN dendrites in lobula and found that they too show increased size, similar to M-DCNs. We have added these findings in **Supplemental Figure S4** of revised manuscript (see figures 5 and 7 above).

7. Please reveal the details/identity of the control KK30B40D.

This line is a correct control line for KK library landing sites at 40D and 30B in KK library host strain. VDRC #60101, line contains Gal4-responsive UAS repeats but no functional RNAi coding sequence at 40D and no transgene insertion at 30B. We mention its identity in Star Methods list.

8. Overexpression of dTzap does not affect mito genes – this is interesting but there are often cases where transcripts can be overexpressed but not proteins – is it possible to show the OE of proteins? If not, authors may include this possibility in Discussion.

We thank reviewer for this point. We have included in the discussion on **page 31** of the revised manuscript.

9. Additional assays are needed to validate aberrant mitochondria functions despite the morphological defects.

We have now shown calcium buffering and ATP synthesis defects (see above).

10. Line 375, “a” factor

This has been fixed.

11. Line 354, if COS7 cells are already stressed and showing phenotypes, it is unclear whether they are good models for additional tests – the final phenotypes could be a synthetic phenotype instead of a mito phenotype. Please clarify.

COS7 cells were used as a first model to check for mitochondrial morphology, as their size gives the opportunity to visualize mitochondria in detail. Then we checked some mitochondria parameters (area, perimeter, aspect ratio and form factor) in hippocampal rat neurons, transfected with shRNA against TZAP. We observe same trend in morphological changes for this model, assuming it is not an artifact of COS7 cells.

12. Figure 6B – seems Drp1 depletion leads to mito phenotypes in current images – maybe another representative image?

We do not detect significant differences in size. The difference in intensity is a normal sample to sample variation after IHC. We changed representative **Figure 6B**.

13. line 442, only half of DCNs innervate – do the other half of DCNs die or stay alive?

According to our MCFO experiment and DCN cell number quantification we do not see signs of cell death in case of Mirana knockdown.

14. Line 459, current data cannot determine the causality between loss of activity vs. connectivity unless loss of activity occurs first. In addition, the loss of postsynaptic sites can be solely caused by loss of presynaptic release – if the synapses are not used, neurons remove them. It is actually surprising that some presynaptic sites are still established and maintained, considering the presynaptic release is blocked.

We do agree this is surprising. One possibility is that the reduction in release allows just enough to maintain presynaptic proteins at their sites. In terms of the direction of causality, we believe that our data in the original manuscript showing that loss of activity is sufficient to abrogate connectivity (using 3 different silencing systems, **now Figures 5 and S7 in the revised manuscript**) support the causality in the indicated direction.

15. Line 783, “25 clusters/sample” means 25 clusters were collected from one section or one animal?

In one animal there are only 2 DCN clusters, that are composed approximately of 60-120 cells/each. Numerous fly heads were frozen in the same block for cryosections. We collected clusters at least from 10-15 different animals per sample.

16. Line 787, please detail how the protocol was adapted.

This kit is adapted for RNA isolation from LCM samples (see RNAqueous-Micro, #1931) instruction manual at page 9. In our isolation procedure we followed manufacturer instructions.

17. Line 799, trimming is needed.

We ran DESeq2 normalization, too the 1000 most variable genes (based on standard deviation) and created an MDS plot. Based on MDS clusters we excluded one DCN sample and re-ran DESeq2.

18. Line 802, please include what kind of scaling analyses were performed to exclude samples
We used log2-scaling

19. Line 825, please indicate what kind of primer sequences were trimmed for sequencing-through reads.

We used standard Illumina adapter trimming using Cutadapt (--max-n 4) and Trimmomatic (TRAILING 1; SLIDINGWINDOW 4 and cut off 20; LEADING 20; MINLEN 50) as stated in the original manuscript.

20. Line 830, typo fond. Plus, phantom peaks have been annotated and identified in black/grey lists for various species. Authors may intersect their peak calls with black/grey lists to identify and remove those phantom peaks. Removing KO peaks is not a standard method to analyze ChIP-seq. KO samples may show peaks if there are residue target proteins.

We corrected the typo. While we agree with the reviewer in cases where de novo data to identify all possible targets is the goal, we note that we were not interested in identifying all peaks of human TZAP. Rather, we focused on whether there were significant peaks around known mitochondrial genes.

21. Line 835, add (Ref).

Reference has been added.

22. Line 874, please add reference or include details about how those programs perform alignment beyond visualization.

Reference has been added.

23. Line 368, it would be helpful if better figure labeling is included to reflect GCaMP.

We have improved figure labeling by adding "GCaMP6f" to the figure panel.

24. In figure 6, some genes are required for mito development but not all genes – how efficient are those RNAi strains? Are they all strong?

While we have not assessed each strain independently, we used strains previously used in the literature to report phenotypes. Furthermore, as we now show, RNAis against genes we now tested (e.g. Drp1, MTFP1 and Parkin) also show phenotypes suggesting they are functional and effective.

25. The causality or the order of phenotypes may not be authors' scope in this paper but may be interesting to the field – which occurs first – mito phenotypes, activity defects and connectivity defects? The order may imply the causality. This is not necessary for revision but may be of authors' interest.

This is a good point. As shown in the original manuscript, the mitochondrial phenotypes appear during development, strongly suggesting they are causal to the release and connectivity phenotypes, which occur later by definition. Furthermore, the rescue by Pink1 strongly supports this direction of causality.

Revision 02.

Reviewer #4 (Remarks to the Author)

My concerns are mostly addressed. I suggest some text rephrasing.

- *Two reviewers raised the same concern about cell-type specificity of the transcriptomic data. If authors choose to emphasize such specificity, strong validation is required to demonstrate the presence of DCN cells and the absence of non-DCN cells. All information authors provided indicates the presence of DCN cells, but no information about markers of non-DCN cells, e.g., glial cells, is presented. The presence of DCN marker genes does not promise the absence of contamination. It is inappropriate to conclude about cell-type specificity without further validation. Indeed, identifying cell type-specific mechanisms is challenging but definitely feasible. FACS has been successfully used to isolate similarly rare neuroblasts (~200 per brain, PMID: 23660757; PMID: 22884370) for multiomics analyses with solid validation of specificity, as well as the extremely rare LNvs neurons (8 per brain, PMID: 30380410) and gamma neurons (PMID: 30300589). If no additional assays are performed, I recommend authors to tune down their conclusions about “cell type-specific” transcriptome.*

We agree with reviewer about possibility of FACS sorting even in case of rare cell types, however in this case reliable and highly specific fluorescence labelling should be used for selected cell types. Unfortunately, in our case we lack specific enough Gal4 driver line, that would be active exclusively in DCNs/LC14 (as it's also expressed in Ventral cluster or VCNs, which quantitatively predominates DCNs). So, in our case we would need to dissect clusters first, and then use dissected tissue for cell sorting, which combined together would be extremely challenging. Also, after laser dissection, dissociation and FACS sorting one could end up with poor quality of RNA samples. We favored LMD in to avoid unspecific FACS, that could include even more contamination to our samples from other cell types. In the table below we extracted DEG of main glial markers, which are decreased in our DCNs sample. Nevertheless, we agree that it is impossible to exclude any contamination from neighboring cells and we will mention this in the manuscript text (Line 104, 108).

Gene	Glial cell type	baseMean	log2FoldChange	lfcSE	stat	pvalue	padj
repo	panglial	313,81751	-1,2598287	2,4550834	-0,5131511	0,6078456	0,9128842
moody	surface glia	319,24028	-1,0036169	2,5478935	-0,3939006	0,6936544	0,9426173
wrapper	cortex glia	1872,5265	1,6434782	1,2445354	1,3205556	0,1866496	0,59453
alm	astrocytes	1418,3212	-1,6689359	1,6191237	-1,0307649	0,3026511	0,7325484
Gat		41886,366	-5,604729	1,0440381	-5,3683183	7,95E-08	4,25E-06
Eaat2	midline glial cells/eye	7816,6302	-1,5620987	0,8398924	-1,8598795	0,0629026	0,3056138

- *Based on supplementary file 1 tab 1, seven KD samples are used for analyses rather than six, line 101.*

We used 6 samples for the analysis, and edited file 1 tab 1.

- *The text states, “Using an endogenously GFP-tagged version to examine...” so it implies that the endogenous Mirana is tagged by GFP. Authors responses suggested that the Mirana-GFP is a transgenic insertion, in conflict with the text. Please clarify and cite the transgenic fly strain.*

GFP tagged line of Mirana (BDRC #67655 is mentioned in Figure 2 legend (line 228). We also edited main text in line 179.

- *Even if the Mirana-GFP is a transgenic insertion, it is important and a great tool to titrate Mirana protein degradation by monitoring GFP levels since RNAi targets both endogenous and transgenic Mirana RNAs. The point is, when does the Mirana protein function during development? RNAi was induced between L3 and APF48 to knockdown RNAs, but protein levels may stay unchanged, therefore it is arguable to conclude about temporal Mirana functions solely based on when RNAs are degraded. This is a great manuscript proposing precise temporal resolution during development. I suggest at least discuss such possible RNA/protein discordance in Discussion if no additional experiments are performed.*

We do not quite understand this argument.

-First, both single cell RNAseq data and Mirana-GFP show that Mirana RNA and protein are temporally expressed. In other words, Mirana function is by definition temporal.

-Second, if one assumes that the RNAi did not reduce Mirana protein levels, one must explain how come phenotypes are observed upon RNAi expression between L3 and P48, but not at later stages. If the phenotypes were somehow an RNAi tool artefact, they should not depend on time of expression. If the temporal RNAi did not degrade the protein, then one must explain how come there are phenotypes present when the RNAi is performed when the protein is there but not when the protein is not there.

-Third, the CRISPR knock-out removes the gene before expression is initiated, and shows the same phenotypes as the RNAi.

The most reasonable explanation for the sum of these observations is the temporal requirement of Mirana in DCNs. Introducing a statement about the caveat proposed by the reviewer requires qualifying the caveat by its own caveats stated above. We think adding all this to the discussion section will be more likely to introduce confusion than clarity to the reader.

- *is it a typo in line 153 “at an earlier developmental stage (P48)” since authors just mentioned earlier that loss of Mirana between L3 and P48 leads to defects, but then P48 has not difference?*

We rephrased the sentence to make it clearer.

- *Please make it clear embryo ChIP-seq was used.*

We used Drosophila whole organism (adult female) ChIP-seq data available in ModEncode <https://www.encodeproject.org/experiments/ENCSR033ZHC>.

We now mention this in line 317 (Supplementary Figure S6 description).

- *I meant reads need to be trimmed for sequencing quality and adapter sequences before alignment for RNA-seq analyses, line 759.*

Reads were trimmed for both sequencing quality and adapter sequences using Cutadapt with the -a parameter for adapter removal and Trimmomatic for quality filtering.

- *This command does not trim adapter sequences, unless -a/-A or -g/-G is used.*

Cutadapt was run with the -a option to explicitly remove adapter sequences. The updated methods now accurately describe both adapter and quality trimming steps.